# Unified lower bounds for interactive high-dimensional estimation under information constraints

**Jayadev Acharya**
Cornell University
acharya@cornell.edu

**Clément L. Canonne**
University of Sydney
clement.canonne@sydney.edu.au

**Ziteng Sun**
Google Research, New York
zitengsun@google.com

**Himanshu Tyagi**
Indian Institute of Science, Bangalore
htyagi@iisc.ac.in

## Abstract

We consider distributed parameter estimation using interactive protocols subject to *local information constraints* such as bandwidth limitations, local differential privacy, and restricted measurements. We provide a unified framework enabling us to derive a variety of (tight) minimax lower bounds for different parametric families of distributions, both continuous and discrete, under any $\ell_p$ loss. Our lower bound framework is versatile and yields "plug-and-play" bounds that are widely applicable to a large range of estimation problems. In particular, our approach recovers bounds obtained using data processing inequalities and Cramér–Rao bounds, two other alternative approaches for proving lower bounds in our setting of interest. Further, for the families considered, we complement our lower bounds with matching upper bounds.

## 1 Introduction

We consider the problem of parameter estimation under *local information constraints*, where the estimation algorithm has access to only limited information about each sample. These constraints can be of various types, including communication constraints, where each sample must be described using a few (*e.g.*, constant number of) bits; (local) privacy constraints, where each sample is obtained from a different user and the users seek to reveal as little as possible about their specific data; as well as many others, *e.g.*, noisy communication channels, or limited types of data access such as linear measurements. Such problems have received significant attention in recent years, motivated by applications such as data analytics in distributed systems and federated learning.

Our main focus is on information-theoretic lower bounds for the minimax error rates (or, equivalently, the sample complexity) of these problems. Several recent works have provided bounds that apply to specific constraints or work for specific parametric estimation problems, sometimes without allowing for interactive protocols. Indeed, handling interactive protocols is technically challenging, and several results in prior work exhibit flaws in their analysis. In particular, even the most basic Gaussian mean estimation problem using interactive communication remains, surprisingly, open.

We present general, "plug-and-play" lower bounds for parametric estimation under information constraints that can be used for any local information constraint and allows for *interactive* protocols. Our abstract bound requires very simple (and natural) assumptions to hold for the underlying parametric family; in particular, we do not require technical "regularity" conditions that are common in asymptotic statistics.

37th Conference on Neural Information Processing Systems (NeurIPS 2023).

We apply our general bound to canonical problems of high-dimensional mean estimation and distribution estimation, under privacy and communication constraints, for the entire family of $\ell_p$ loss functions for $p \geq 1$. In addition, we provide complementary schemes that show that our lower bounds are tight for most settings of interest.

## 1.1 Our results

Our main contribution is a general approach to establish lower bounds in distributed information-constrained parameter estimation. The setup is described in detail in Section 2 and is illustrated in Fig. 1. In short, independent samples $X^n = (X_1, \ldots, X_n)$ are generated from an unknown distribution $\mathbf{p}$ from a parametric family $\mathcal{P}_\Theta = \{\mathbf{p}_\theta, \theta \in \Theta\}$ of distributions. Only limited information $Y_i$ about datum $X_i$ is available to the algorithm. The goal is to estimate the underlying parameter $\theta$ associated with $\mathbf{p}$. Furthermore, we consider interactive estimation, wherein $Y_i$ can depend on $Y_1, \ldots, Y_{i-1}$. Our general lower bound, which we develop in Section 3, takes the following form: Consider a collection of distributions $\{\mathbf{p}_z\}_{z \in \{-1,+1\}^k} \subseteq \mathcal{P}_\Theta$ contained in the parametric family. This collection represents a "difficult subproblem" that underlies the parametric estimation problem being considered; such constructions are often used when deriving information-theoretic lower bounds (*e.g.*, in Assouad's method [33]). Note that each coordinate of $z$ represents, in essence, a different "direction" of uncertainty for the parameter space. The difficulty of the estimation problem can be related to the difficulty of determining a randomly chosen $z$ (or most of the coordinates of $z$), denoted $Z$, by observing samples from $\mathbf{p}_z$. Once $Z = z$ is fixed, $n$ independent samples $X^n = (X_1, \ldots, X_n)$ are generated from $\mathbf{p}_z$ and the limited information $Y_i$ about $X_i$ is passed to an estimator.

Our most general result, stated as Theorem 1, is an upper bound for the average discrepancy, an average distance quantity related to average probability of error in determining coordinates of $Z$ by observing the limited information $Y^n = (Y_1, \ldots, Y_n)$. Our bounding term reflects the underlying information constraints using a quantity that captures how "aligned" we can make our information about the sample to the uncertainty in different coordinates of $Z$. Importantly, our results hold under minimal assumptions. In particular, in contrast to many previous works, our results do not require any "bounded ratio" assumption on the collection $\{\mathbf{p}_z\}_{z \in \{-1,+1\}^k}$, which would ask that the density function change by at most a constant factor if we modify one coordinate of $z$. When we impose additional structure for $\mathbf{p}_z$ – such as orthogonality of the random changes in density when we modify different coordinates of $z$ and, more stringently, independence and subgaussianity of these changes – we get concrete bounds which are readily applicable to different problems. These plug-and-play bounds are stated as consequences of our main result in Theorem 2. The interested reader can also directly consider the applications to local privacy (Corollary 1) or communication constraints (Corollary 2).

We demonstrate the versatility of the framework by showing that it readily yields tight (and in some cases nearly tight) bounds for parameter estimation (both in the sparse and dense cases) for several fundamental families of continuous and discrete distributions, several families of information constraints such as communication and local differential privacy (LDP), and for the family of $\ell_p$ loss functions for $p \geq 1$, all when interactive protocols are allowed. To complement our lower bounds, we provide algorithms (protocols) which attain the stated rates, thus establishing optimality of our results.[1] We discuss these results in Section 5, where we provide the corresponding statements. In terms of the applications, our contributions are two-fold:

1. We obtain several results from a diverse set of prior works in a unified fashion as simple corollaries of our main result: As discussed further in Section 1.2, the lower bounds for mean estimation for product Bernoulli under $\ell_2$ loss and those for estimation of discrete distributions under $\ell_1$ and $\ell_2$ losses, for both communication and LDP constraints, were known from previous work. However, our approach allows us to easily recover those results and extend them to arbitrary $\ell_p$ losses, with interaction allowed, in a unified fashion.

2. Our bounds also yield new lower bounds for some canonical problems. The prototypical example being mean estimation for high-dimensional Gaussian distributions under information constraints. As discussed in the next section, while some prior work claimed lower bounds for this problem, their arguments appear to be flawed – at a high level, due to the

---

[1] Up to a logarithmic factor in the case of $\ell_\infty$ loss, or, for some of our bounds, with a mild restriction on $n$ being large enough.

"bounded ratio" assumption their techniques rely on, which Gaussian distributions do not satisfy, and which our framework does not require. To the best of our knowledge our work is the first to obtain those lower bounds for interactive mean estimation of high-dimensional Gaussian distributions under communication or local privacy constraints.

## 1.2 Previous and related work

There is a significant amount of work in the literature dedicated to parameter estimation under various constraints and settings. Here, we restrict our discussion to works that are most relevant to the current paper, with a focus on the interactive setting (either the sequential or blackboard model; see Section 2 for definitions).

The work arguably closest to ours is the recent work [5], which focuses on density estimation and goodness-of-fit testing of discrete distributions, under the $\ell_1$ metric, for sequentially interactive protocols under general local information constraints (including, as special cases, local privacy and communication constraints, as in the present paper). This work can be seen as a significant generalization of the techniques of [5], allowing us to obtain lower bounds for estimation in a variety of settings, notably high-dimensional parameter estimation.

Among other works on high-dimensional mean estimation under communication constraints, [17, 10] consider communication-constrained Gaussian mean estimation in the blackboard communication model, under $\ell_2$ loss. The protocols for the upper bounds in these works do not require interactivity and are complemented with lower bounds which show that the bounds are tight up to constant factors in the dense case and up to logarithmic factors in the sparse case. However, the proof of the lower bound in [10] seems to present a gap (specifically, in the truncation argument of [10, Theorem 4.3]), as confirmed in personal communication with the authors. Correcting the issue in the truncation argument would lead to a result significantly weaker than the claimed lower bound, and it is unclear whether this can be fixed using the techniques from that paper.

In this work, we present interactive protocols for the sparse case which improve over the noninteractive protocols and strenghten the upper bounds by a logarithmic factor in the interactive case (see Remark 1). Further, using our general framework, we establish a nearly-matching lower bound for the problem, recovering the rate lower bound originally claimed in [10] up to a logarithmic factor. In a slightly different setting, [26] considers the mean estimation problem for product Bernoulli distributions when the mean vector is 1-sparse, under $\ell_2$ loss. The lower bound in [26], too, allows sequentially interactive protocols. In the blackboard communication model, [19] and [18] obtained tight bounds for mean estimation and density estimation under $\ell_2$ and $\ell_1$ loss, respectively.

Turning to local privacy, [23] provide upper bounds (as well as some partial lower bounds) for one-dimensional Gaussian mean estimation under LDP under the $\ell_2$ loss, in the sequentially interactive model. Recent works of [7] and [8] obtain lower bounds for mean estimation in the blackboard communication model and under LDP, respectively, for both Gaussian and product Bernoulli distributions; as well as density estimation for discrete distributions. Their approach is based on the classic Cramér–Rao bound and, as such, is tied inherently to the use of the $\ell_2$ loss. In a recent independent work, [25] extended these methods to obtain lower bounds under general $\ell_p$ loss under communication constraints, which are tight for Gaussian mean estimation under noninteractive protocols. [14], by developing a locally private counterpart of some of the well-known information-theoretic tools for establishing statistical lower bounds (namely, Le Cam, Fano, and Assouad), establish tight or nearly tight bounds for several mean estimation problems in the LDP setting.

More recently, drawing on machinery from the communication complexity literature, [13] develop a methodology for proving lower bounds under LDP constraints in the blackboard communication model. They obtain lower bounds for mean estimation of product Bernoulli distributions under general $\ell_p$ losses which match ours (in the high-privacy regime, *i.e.*, small $\varepsilon$). Similar to the results under communication constraints [17, 10], their approach relies heavily on the assumption that the distributions on each coordinate are independent, which fails to generalize to discrete distributions. Further, their bounds are tailored to the LDP constraints and do not seem to extend to arbitrary information constraints. Finally, while [13] also claims tight bounds for mean estimation of Gaussian and sparse Gaussian distributions under $\ell_2$ loss, their argument invokes the analogous (flawed) result from [10] and thus it is unclear whether the stated lower bound can be shown using their techniques.

Finally, we mention that very recently, following the appearance of [5], an updated version of [19] appeared online as [21], which has similar results as ours for the high-dimensional mean estimation problem under communication constraints. Both our work and [21] build upon the framework presented for the discrete setting in [5]. Moreover, their work still need the "bounded ratio", and hence their lower bound for sparse Gaussian family only works for noninteractive protocols.

**Notation.** Hereafter, we write $\log$ and $\ln$ for the binary and natural logarithms, respectively. For distributions $\mathbf{p}_1, \mathbf{p}_2$ over $\mathcal{X}$, denote their Kullback–Leibler divergence (in nats) by $\mathrm{D}(\mathbf{p}_1\|\mathbf{p}_2)$, and their Hellinger distance by $\mathrm{d}_{\mathrm{H}}(\mathbf{p}_1, \mathbf{p}_2) := (\frac{1}{2}\int(\sqrt{\frac{\mathrm{d}\mathbf{p}_1}{\mathrm{d}\lambda}} - \sqrt{\frac{\mathrm{d}\mathbf{p}_2}{\mathrm{d}\lambda}})^2 \, \mathrm{d}\lambda)^{1/2}$, where we assume $\mathbf{p}_1, \mathbf{p}_2 \ll \lambda$ for some underlying measure $\lambda$ on $\mathcal{X}$. Further, we denote the Shannon entropy of a random variable $X$ by $H(X)$ and the mutual information between $X$ and $Y$ by $I(X;Y)$; we will sometimes write $H(\mathbf{p})$ for the entropy of a random variable with distribution $\mathbf{p}$. We refer the reader to [12] for details on these notions and their properties, which will be used throughout. Given two functions $f, g$, we write $f \lesssim g$ if there exists an absolute constant $C > 0$ such that $f(x) \leq Cg(x)$ for all $x$; and $f \asymp g$ if $f \lesssim g$ and $f \gtrsim g$ both hold. Finally, we use the standard asymptotic notation $O(f), \Omega(f), \Theta(f)$.

**Organization.** In Section 2, we formalize our setting of interactive inference under local information constraints. The general lower bound framework and results are presented in Section 3, where we provide implications of the general result under additional structures and specific information constraints such as local privacy (LDP) and communication constraints. Finally, we use our framework to readily derive lower bounds for a wide range of applications in Section 5. Due to space constraints, all proofs, as well as our upper bounds (algorithms) are provided in the Supplement, where we also discuss how our techniques compare with other existing approaches for proving lower bounds under information constraints, namely, those based on strong data processing inequalities (SDPI) or on the van Trees inequality.

## 2 The setup

We consider standard parametric estimation problems. For some $\Theta \subseteq \mathbb{R}^d$, let $\mathcal{P}_\Theta = \{\mathbf{p}_\theta, \theta \in \Theta\}$ be a family of distributions over some measurable space $(\mathcal{X}, \mathfrak{X})$, namely each $\mathbf{p}_\theta$ is a distribution over $(\mathcal{X}, \mathfrak{X})$. Suppose $n$ independent samples $X^n = (X_1, \ldots, X_n)$ from an unknown $\mathbf{p}_\theta \in \mathcal{P}_\Theta$ are obtained. The goal in parametric estimation is to design estimators $\hat{\theta} : \mathcal{X}^n \to \Theta$, and form estimates $\hat{\theta}(X^n)$ of $\theta$ using independent samples $X^n$ from $\mathbf{p}_\theta$. We illustrate our results using two specific distribution families: discrete probability mass functions (pmfs) and high-dimensional product distributions with unknown mean vectors. We will describe the precise minimax setting in detail later in this section.

We are interested in an information-constrained setting, where we do not have direct access to the samples $X^n$ from $\mathbf{p}_\theta$. Instead, we can only obtain limited information about each datapoint $X_i$. Following [3], we model these information constraints by specifying an allowed set of *channels* $\mathcal{W}$ with input alphabet $\mathcal{X}$ and some output space $\mathcal{Y}$.[2] Each sample $X_i$ is passed through a channel from $\mathcal{W}$, chosen appropriately, and its output $Y_i$ is the observation we get. This setting is quite general and captures as special cases the popular communication and privacy constraints, as we will describe momentarily.

We now formally describe the setting, which is illustrated in Fig. 1. $n$ i.i.d. samples $X_1, \ldots, X_n$ from an unknown distribution $\mathbf{p}_\theta \in \mathcal{P}_\Theta$ are observed by players (users) where player $t$ observes $X_t$. Player $t \in [n]$ selects a channel $W_t \in \mathcal{W}$ and sends the message $Y_t$ to a referee, where $Y_t$ is drawn from the probability measure $W_t(\cdot \mid X_t, Y_1, \ldots, Y_{t-1})$. The referee observes $Y^n := (Y_1, \ldots, Y_n)$ and seeks to estimate the parameter $\theta$.

The freedom allowed in the choice of $W_t$ at the players gives rise to various communication protocols. We focus on *interactive protocols*, where channels are chosen by one player at a time, and they can use all previous messages to make this choice. We describe this class of protocols below, where we further allow each player $t$ to have a different set of constraints $\mathcal{W}_t$ (*e.g.*, a different communication budget), and $W_t$ must be in $\mathcal{W}_t$. For simplicity of exposition, and as these already encapsulate most

---

[2]Formally, a channel is a Markov kernel $W : \mathfrak{Y} \times \mathcal{X} \to [0, 1]$, which we assume to be absolutely continuous with respect to some underlying measure $\mu$ on $(\mathcal{Y}, \mathfrak{Y})$. When clear from the context, we will drop the reference to the $\sigma$-algebras $\mathfrak{X}$ and $\mathfrak{Y}$; in particular, in the case of finite $\mathcal{X}$ or $\mathcal{Y}$.

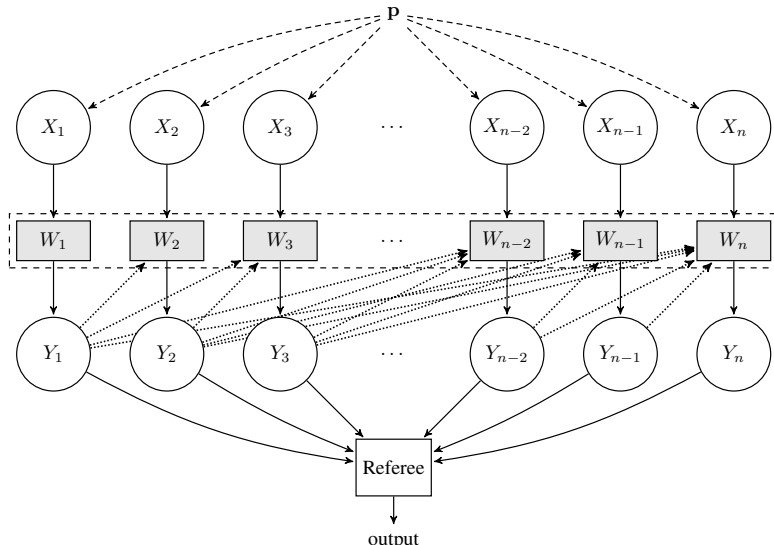

Figure 1: The information-constrained distributed model. In the interactive setting, $W_t$ can depend on the previous messages $Y_1, \ldots, Y_{t-1}$ (dotted, upwards arrows).

of the difficulties, we focus here on the case of *sequentially* interactive protocols, which has been widely considered in the literature and captures many settings of interest. However, we emphasize that our results extend to the more general class of fully interactive protocols (see Supplement).

*Definition* 1 (Sequentially Interactive Protocols). Let $X_1, \ldots, X_n$ be i.i.d. samples from $\mathbf{p}_\theta$, $\theta \in \Theta$. A *sequentially interactive protocol* $\Pi$ *using* $\mathcal{W}^n = (\mathcal{W}_1, \ldots, \mathcal{W}_n)$ involves mutually independent random variables $U, U_1, \ldots, U_n$ (independent of the input $X_1, \ldots, X_n$) and mappings $g_t \colon (U, U_t) \mapsto W_t \in \mathcal{W}_t$ for selecting the channel in round $t \in [n]$. In round $t$, player $t$ uses the channel $W_t$ to produce the message (output) $Y_t$ according to the probability distribution $W_t(\cdot \mid X_t, Y_1, \ldots, Y_{t-1})$. The messages $Y^n = (Y_1, \ldots, Y_n)$ received by the referee and the public randomness $U$ (available to all players) constitute the *transcript* of the protocol $\Pi$; the private randomness $U_1, \ldots, U_n$ (where $U_t$ is local to player $t$) is not part of the transcript. In other words, the channel at player $t$ as a (randomized) mapping $W \colon \mathcal{X} \times \mathcal{Y}^{t-1} \to \mathcal{Y}$, which depends on input $x \in \mathcal{X}$ and the previous $t-1$ messages $y^{t-1} \in \mathcal{Y}^{t-1}$ outputs some $y \in \mathcal{Y}$.

For concreteness, we now instantiate this definition for the two aforementioned types of information constraints, communication and (local) privacy.

**Communication constraints** Let $\mathcal{Y} := \{0,1\}^* = \bigcup_{m=0}^\infty \{0,1\}^m$. For $\ell \geq 1$ and $t \geq 1$, let

$$\mathcal{W}^{\mathrm{comm},\ell} := \{W \colon \mathcal{X} \times \mathcal{Y}^* \to \{0,1\}^\ell\} \tag{1}$$

be the family of channels with input alphabet $\mathcal{X}$ and output alphabet the set of all $\ell$-bit strings. This captures the constraint where the message from each player can be at most $\ell$ bits long, and corresponds to the choice $\mathcal{W}^n = (\mathcal{W}^{\mathrm{comm},\ell}, \ldots, \mathcal{W}^{\mathrm{comm},\ell})$. Note that allowing a different communication budget to each player can be done by setting $\mathcal{W}^n = (\mathcal{W}^{\mathrm{comm},\ell_1}, \ldots, \mathcal{W}^{\mathrm{comm},\ell_n})$.

**Local differential privacy constraints** For $\varepsilon > 0$ and $t \geq 1$, a channel $W \colon \mathcal{X} \times \mathcal{Y}^{t-1} \to \mathcal{Y}$ is $\varepsilon$-*locally differentially private (LDP)* [16, 15, 24] if

$$\sup_{S \in \mathfrak{Y}} \sup_{y^{t-1} \in \mathcal{Y}^{t-1}} \frac{W(S \mid x_1, y^{t-1})}{W(S \mid x_2, y^{t-1})} \leq e^\varepsilon, \quad \forall x_1, x_2 \in \mathcal{X}. \tag{2}$$

We denote by $\mathcal{W}^{\mathrm{priv},\varepsilon}$ the set of all $\varepsilon$-LDP channels. For sequentially interactive protocols, the $\varepsilon$-LDP condition is captured by setting $\mathcal{W}^n = (\mathcal{W}^{\mathrm{priv},\varepsilon}, \ldots, \mathcal{W}^{\mathrm{priv},\varepsilon})$. As before, one can allow different privacy parameters for each player by setting $\mathcal{W}^n = (\mathcal{W}^{\mathrm{priv},\varepsilon_1}, \ldots, \mathcal{W}^{\mathrm{priv},\varepsilon_n})$.

Finally, we formalize the interactive parametric estimation problem for the family $\mathcal{P}_\Theta = \{\mathbf{p}_\theta, \theta \in \Theta\}$. We consider the problem of estimating $\theta$ under $\ell_p$ loss. For $p \in [1, \infty)$, the $\ell_p$ distance between

$u, v \in \mathbb{R}^d$ is $\ell_p(u, v) = \|u - v\|_p = \left( \sum_{i=1}^d |u_i - v_i|^p \right)^{1/p}$. This definition extends in a natural way to $p = \infty$ by taking the limit.[3]

*Definition* 2 (Sequentially Interactive Estimates). Fix $d \in \mathbb{N}$ and $p \in [0, \infty]$. Given a family $\mathcal{P}_\Theta$ of distributions on $\mathcal{X}$, with $\Theta \subset \mathbb{R}^d$, an *estimate* for $\mathcal{P}_\Theta$ consists of a sequentially interactive protocol $\Pi$ with transcript $(Y^n, U)$ and estimator $\hat{\theta} \colon (Y^n, U) \mapsto \hat{\theta}(Y^n, U) \in \Theta$. The referee observes the transcript $(Y^n, U)$ and forms the estimate $\hat{\theta}(Y^n, U)$ of the unknown $\theta$. Further, for $n \in \mathbb{N}$ and $\gamma \in (0, 1)$, $(\Pi, \hat{\theta})$ constitutes an $(n, \gamma)$-*estimator* for $\mathcal{P}_\Theta$ using $\mathcal{W}$ under $\ell_p$ loss if for every $\theta \in \Theta$ the transcript $(Y^n, U)$ of $\Pi$ satisfies

$$\mathbb{E}_{\mathbf{p}_\theta^n} \left[ \ell_p(\theta, \hat{\theta}(Y^n, U))^p \right]^{1/p} \leq \gamma.$$

Note that the expectation is over the input $X^n \sim \mathbf{p}_\theta^n$ for the protocol $\Pi$ and the randomness of $\Pi$.

## 3 Main result: The information contraction bound

Our main result is a unified framework to bound the information revealed about the unknown $\theta$ by the transcript of the messages obtained via the constraints defined by the channel family $\mathcal{W}$. The framework is versatile and provides tight bounds for several families of continuous and discrete distributions, several families of information constraints such as communication and local differential privacy, and for the family of $\ell_p$ loss functions for $p \geq 1$.

Our approach at a high-level proceeds as below: We first consider the "pertubation space" $\mathcal{Z} := \{-1, +1\}^k$, for some suitable $k$. We associate with each $z \in \mathcal{Z}$ a parameter $\theta_z \in \Theta$, and refer to $\mathbf{p}_{\theta_z}$ simply as $\mathbf{p}_z$. These distributions are designed in a way that the distance between $\theta_z$ and $\theta_{z'}$ is large when the Hamming distance between $z$ and $z'$ is large. With this, the difficulty of estimating $\theta$ will be captured in the difficulty of estimating the associated $z$. This will make our approach compatible with the standard Assouad's method for deriving lower bounds (*cf.* [33]).

Then, we let $Z = (Z_1, \ldots, Z_k)$ be a random variable over $\mathcal{Z}$. Under some assumptions on the distribution of $Z$, we will bound the information between the individual $Z_i$s and the transcript $(Y^n, U)$ induced by a family of channels $\mathcal{W}$. Combining the two steps above provides us with the desired lower bounds. Formally, let $\mathcal{Z} := \{-1, +1\}^k$ for some $k$ and $\{\mathbf{p}_z\}_{z \in \mathcal{Z}}$ (where $\mathbf{p}_z = \mathbf{p}_{\theta_z}$) be a collection of distributions over $\mathcal{X}$, indexed by $z \in \mathcal{Z}$. For $z \in \mathcal{Z}$, denote by $z^{\oplus i} \in \mathcal{Z}$ the vector obtained by flipping the sign of the $i$th coordinate of $z$. To bound the information that can be obtained about the underlying $z$ from the observations, we make the following assumptions:

**Assumption 1** (Densities Exist). *For every $z \in \mathcal{Z}$ and $i \in [k]$ it holds that $\mathbf{p}_{z^{\oplus i}} \ll \mathbf{p}_z$, and there exist measurable functions $\phi_{z,i} \colon \mathcal{X} \to \mathbb{R}$ such that $\frac{d\mathbf{p}_{z^{\oplus i}}}{d\mathbf{p}_z} = 1 + \phi_{z,i}$.*

The functions $\phi_{z,i}$ capture the change in density when the coordinate $i$ is flipped. In our applications below, we will have discrete distributions or continuous densities, and the Radon–Nikodym derivatives above can be replaced with the corresponding ratios between the pmfs and pdfs, respectively.

**Assumption 2** (Orthogonality). *There exists some $\alpha^2 \geq 0$ such that, for all $z \in \mathcal{Z}$ and distinct $i, j \in [k]$, $\mathbb{E}_{\mathbf{p}_z}[\phi_{z,i} \phi_{z,j}] = 0$ and $\mathbb{E}_{\mathbf{p}_z}[\phi_{z,i}^2] \leq \alpha^2$.*

Note that from Assumption 1 we have that $\mathbb{E}_{\mathbf{p}_z}[\phi_{z,i}] = 0$ for each $i$. In conjunction with Assumption 2 this implies that for any fixed $z \in \mathcal{Z}$, the family $(1, \phi_{z,1}, \ldots, \phi_{z,k})$ is orthogonal and uniformly bounded in $L^2(\mathcal{X}, \mathbf{p}_z)$. Taken together, Assumption 1 and Assumption 2 roughly say that the densities can be decomposed into uncorrelated "perturbations" across coordinates of $\mathcal{Z}$. In later sections, we will show that for several families, such as discrete distributions, product Bernoulli distributions, and spherical Gaussians, well-known constructions for lower bounds satisfy these assumptions.

Our first bound given in (3) only requires Assumption 1; by imposing the additional structure of Assumption 2, we obtain the more specialized bound given in (4). Interestingly, (3) can be strengthened further when the following subgaussianity assumption holds.

---

[3] By Hölder's inequality, we also have $\ell_\infty(u, v) \leq \ell_p(u, v) \leq d^{1/p} \ell_\infty(u, v)$ for all $p \geq 1$ and $u, v \in \mathbb{R}^d$, which implies that for $p := \log d$ we have $\ell_\infty(u, v) \leq \ell_p(u, v) \leq 2\ell_\infty(u, v)$: i.e., $\ell_{\log d}(u, v)$ gives a factor-2 approximation of the $\ell_\infty$ loss. This further extends to $s$-sparse vectors, with a factor $s^{1/p}$ instead of $d^{1/p}$.

**Assumption 3** (Subgaussianity). *There exists some $\sigma \geq 0$ such that, for all $z \in \mathcal{Z}$, the random vector $\phi_z(X) := (\phi_{z,i}(X))_{i\in[k]} \in \mathbb{R}^k$ is $\sigma^2$-subgaussian for $X \sim \mathbf{p}_z$.*[4]

Let $Z = (Z_1, \ldots, Z_k)$ be a random variable over $\mathcal{Z}$ such that $\Pr[\,Z_i = 1\,] = \tau$ for all $i \in [k]$ and the $Z_i$s are all independent; we denote this distribution by $\mathrm{Rad}(\tau)^{\otimes k}$. Our main result is an upper bound on the average amount of information that can be obtained about a coordinate of $Z$ from the transcript $(Y^n, U)$ of a sequentially interactive protocol, as a function of the information constraint channels and $\phi_{Z,i}$s. This result only requires Assumption 1, and is presented below in its most general form, suited to applications beyond those discussed in the current paper.

**Theorem 1** (Information contraction bound: Technical form). *Fix $\tau \in (0, 1/2]$. Let $\Pi$ be a sequentially interactive protocol using $\mathcal{W}^n$, and let $Z$ be a random variable on $\mathcal{Z}$ with distribution $\mathrm{Rad}(\tau)^{\otimes k}$. Let $(Y^n, U)$ be the transcript of $\Pi$ when the input $X_1, \ldots, X_n$ is i.i.d. with common distribution $\mathbf{p}_Z$, with density function $\mathbf{p}_Z^{Y^n}$. Then, under Assumption 1,*

$$\left(\frac{1}{k}\sum_{i=1}^{k} \mathrm{d}_{\mathrm{TV}}\left(\mathbf{p}_{+i}^{Y^n}, \mathbf{p}_{-i}^{Y^n}\right)\right)^2 \leq \frac{7}{k}\sum_{t=1}^{n}\max_{z\in\mathcal{Z}}\max_{W\in\mathcal{W}_t}\sum_{i=1}^{k}\int_{\mathcal{Y}}\frac{\mathbb{E}_{\mathbf{p}_z}[\phi_{z,i}(X)W(y\mid X)]^2}{\mathbb{E}_{\mathbf{p}_z}[W(y\mid X)]}\,\mathrm{d}\mu\,, \quad (3)$$

*where $\mathbf{p}_{+i}^{Y^n} := \mathbb{E}\big[\mathbf{p}_Z^{Y^n} \mid Z_i = +1\big]$, $\mathbf{p}_{-i}^{Y^n} := \mathbb{E}\big[\mathbf{p}_Z^{Y^n} \mid Z_i = -1\big]$.*

We now instantiate this result, invoking Assumptions 2 and 3, to give simple "plug-and-play" bounds which can be applied readily to several inference problems and information constraints.

**Theorem 2.** *Fix $\tau \in (0, 1/2]$. Let $\Pi$ be a sequentially interactive protocol using $\mathcal{W}^n$, and let $Z$ be a random variable on $\mathcal{Z}$ with distribution $\mathrm{Rad}(\tau)^{\otimes k}$. Let $(Y^n, U)$ be the transcript of $\Pi$ when the input $X_1, \ldots, X_n$ is i.i.d. with common distribution $\mathbf{p}_Z$. Then, under Assumptions 1 and 2, we have*

$$\left(\frac{1}{k}\sum_{i=1}^{k} \mathrm{d}_{\mathrm{TV}}\left(\mathbf{p}_{+i}^{Y^n}, \mathbf{p}_{-i}^{Y^n}\right)\right)^2 \leq \frac{7}{k}\alpha^2\sum_{t=1}^{n}\max_{z\in\mathcal{Z}}\max_{W\in\mathcal{W}_t}\int_{\mathcal{Y}}\frac{\mathrm{Var}_{\mathbf{p}_z}[W(y\mid X)]}{\mathbb{E}_{\mathbf{p}_z}[W(y\mid X)]}\,\mathrm{d}\mu\,. \quad (4)$$

*Moreover, if Assumption 3 holds as well, we have*

$$\left(\frac{1}{k}\sum_{i=1}^{k} \mathrm{d}_{\mathrm{TV}}\left(\mathbf{p}_{+i}^{Y^n}, \mathbf{p}_{-i}^{Y^n}\right)\right)^2 \leq \frac{14}{k}\sigma^2\sum_{t=1}^{n}\max_{z\in\mathcal{Z}}\max_{W\in\mathcal{W}_t} I(\mathbf{p}_z; W), \quad (5)$$

*where $I(\mathbf{p}_z; W)$ denotes the mutual information $I(X; Y)$ between the input $X \sim \mathbf{p}_z$ and the output $Y$ of the channel $W$ with $X$ as input.*

As an illustrative and important corollary, we now derive the implications of this theorem for communication and privacy constraints. For both constraints our tight (or nearly tight) bounds in Section 5 follow directly from these corollaries.

**Corollary 1** (Local privacy constraints). *For $\mathcal{W} = \mathcal{W}^{\mathrm{priv},\varepsilon}$ and any family of distributions $\{\mathbf{p}_z, z \in \{-1, +1\}^k\}$ satisfying Assumptions 1 and 2, with the notation of Theorem 2, we have*

$$\left(\frac{1}{k}\sum_{i=1}^{k} \mathrm{d}_{\mathrm{TV}}\left(\mathbf{p}_{+i}^{Y^n}, \mathbf{p}_{-i}^{Y^n}\right)\right)^2 \leq \frac{7}{k}n\alpha^2\big((e^\varepsilon - 1)^2 \wedge e^\varepsilon\big). \quad (6)$$

*Moreover, if Assumption 3 holds as well, we have*

$$\left(\frac{1}{k}\sum_{i=1}^{k} \mathrm{d}_{\mathrm{TV}}\left(\mathbf{p}_{+i}^{Y^n}, \mathbf{p}_{-i}^{Y^n}\right)\right)^2 \leq \frac{14}{k}n\sigma^2\varepsilon. \quad (7)$$

**Corollary 2** (Communication constraints). *For any family of channels $\mathcal{W}$ with finite output space $\mathcal{Y}$ and any family of distributions $\{\mathbf{p}_z, z \in \{-1, +1\}^k\}$ satisfying Assumptions 1 and 2, with the notation of Theorem 2, we have*

$$\left(\frac{1}{k}\sum_{i=1}^{k} \mathrm{d}_{\mathrm{TV}}\left(\mathbf{p}_{+i}^{Y^n}, \mathbf{p}_{-i}^{Y^n}\right)\right)^2 \leq \frac{7}{k}n\alpha^2|\mathcal{Y}|. \quad (8)$$

---

[4]Recall that a r.v. $Y$ is $\sigma^2$-subgaussian if $\mathbb{E}[Y] = 0$ and $\mathbb{E}[e^{\lambda Y}] \leq e^{\sigma^2\lambda^2/2}$ for all $\lambda \in \mathbb{R}$; and that a vector-valued r.v. $Y$ is $\sigma^2$-subgaussian if its projection $\langle Y, v \rangle$ is $\sigma^2$-subgaussian for every unit vector $v$.

*Moreover, if Assumption 3 holds as well, we have*

$$\left( \frac{1}{k} \sum_{i=1}^{k} \mathrm{d}_{\mathrm{TV}} \left( \mathbf{p}_{+i}^{Y^n}, \mathbf{p}_{-i}^{Y^n} \right) \right)^2 \leq \frac{14}{k} n \sigma^2 \log |\mathcal{Y}|. \tag{9}$$

## 4 An Assouad-type bound

In the previous section we provided an upper bound on $\frac{1}{k} \sum_{i=1}^{k} \mathrm{d}_{\mathrm{TV}} \left( \mathbf{p}_{+i}^{Y^n}, \mathbf{p}_{-i}^{Y^n} \right)$. We now prove a lower bound for this quantity in terms of the parameter estimation task we set out to solve. This is an "Assouad's lemma-type" bound, which when combined with Theorem 2 will establish the bounds for $n$; and, reorganizing, the minimax rate lower bounds. To state the result, we require the following assumption, which relates the $\ell_p$ distance between parameters $\theta_z$s to the distance between $z$s.

**Assumption 4** (Additive loss). *Fix $p \in [1, \infty)$. For every $z, z' \in \mathcal{Z} \subset \{-1, +1\}^k$,*

$$\ell_p(\theta_z, \theta_{z'}) = 4\gamma \left( \frac{\mathrm{d}_{\mathrm{Ham}}(z, z')}{\tau k} \right)^{1/p},$$

*where $\mathrm{d}_{\mathrm{Ham}}(z, z') := \sum_{i=1}^{k} \mathbb{1}\{z_i \neq z_i'\}$ denotes the Hamming distance.*

**Lemma 1** (Assouad-type bound). *Let $p \geq 1$ and assume that $\{\mathbf{p}_z, z \in \mathcal{Z}\}$, $\tau \in [0, 1/2]$ satisfy Assumption 4. Let $Z$ be a random variable on $\mathcal{Z} = \{-1, +1\}^k$ with distribution $\mathrm{Rad}(\tau)^{\otimes k}$. Suppose that $(\Pi, \hat{\theta})$ constitutes an $(n, \gamma)$-estimator of $\mathcal{P}_\Theta$ using $\mathcal{W}^n$ under $\ell_p$ loss (see Definition 2) and $\mathrm{Pr}_Z[\mathbf{p}_Z \in \mathcal{P}_\Theta] \geq 1 - \tau/4$. Then the transcript $(Y^n, U)$ of $\Pi$ satisfies*

$$\frac{1}{k} \sum_{i=1}^{k} \mathrm{d}_{\mathrm{TV}} \left( \mathbf{p}_{+i}^{Y^n}, \mathbf{p}_{-i}^{Y^n} \right) \geq \frac{1}{4},$$

*where $\mathbf{p}_{+i}^{Y^n} := \mathbb{E}[\mathbf{p}_Z^{Y^n} \mid Z_i = +1]$, $\mathbf{p}_{-i}^{Y^n} := \mathbb{E}[\mathbf{p}_Z^{Y^n} \mid Z_i = -1]$.*

## 5 Applications

We now consider three distribution families: product Bernoulli distributions and Gaussian distributions with identity covariance matrix (and $s$-sparse mean vectors), and discrete distributions (multinomials), to illustrate the generality and efficacy of our bounds. We describe these three families below, before addressing each of them in their respective subsection. Due to space constraints, we present the proof of the upper bounds in Appendix C, and proof of the lower bounds in Appendix G.

**Sparse Product Bernoulli** ($\mathcal{B}_{d,s}$). Let $1 \leq s \leq d$, $\Theta = \{ \theta \in [-1, 1]^d : \|\theta\|_0 \leq s \}$, and $\mathcal{X} = \{-1, 1\}^d$. Let $\mathcal{P}_\Theta := \mathcal{B}_{d,s}$ be the family of $d$-dimensional $s$-sparse product Bernoulli distributions over $\mathcal{X}$. Namely, for $\theta = (\theta_1, \ldots, \theta_d) \in \Theta$, the distribution $\mathbf{p}_\theta$ is equal to $\otimes_{j=1}^{d} \mathrm{Rad}(\frac{1}{2}(\theta_j + 1))$: a distribution on $\{-1, +1\}^d$ such that the marginal distributions are independent, and for which the mean of the $j$th marginal is $\theta_j$.

**Sparse Gaussian** ($\mathcal{G}_{d,s}$). Let $1 \leq s \leq d$, $\Theta = \{ \theta \in [-1, 1]^d : \|\theta\|_0 \leq s \}$, and $\mathcal{X} = \mathbb{R}^d$. Let $\mathcal{P}_\Theta := \mathcal{G}_{d,s}$ be the family of $d$-dimensional spherical Gaussian distributions with bounded $s$-sparse mean. That is, for $\theta \in \Theta$, $\mathbf{p}_\theta = \mathcal{G}(\theta, \mathbb{I})$ with mean $\theta$ and covariance matrix $\mathbb{I}$. We note that this general formulation assumes $\|\theta\|_\infty \leq 1$ (from the choice of $\Theta$).[5]

**Discrete distributions** ($\Delta_d$). Let $\Theta = \{ \theta \in [0, 1]^d : \sum_{i=1}^{d} \theta_i = 1 \} \subseteq \mathbb{R}^d$ and $\mathcal{X} = \{1, 2, \ldots, d\}$. Let $\mathcal{P}_\Theta := \Delta_d$, where $\Delta_d$ is the standard $(d-1)$-simplex of all probability mass functions over $\mathcal{X}$. Namely, the distribution $\mathbf{p}_\theta$ is a distribution on $\mathcal{X}$, where, for $j \in [d]$, the probability assigned to the element $j$ is $\mathbf{p}_\theta(j) = \theta_j$. For a unified presentation, we view $\theta$ as the mean vector of the "categorical distribution," namely the distribution of vector $(\mathbb{1}\{X = x\}, x \in \mathcal{X})$ for $X$ with distribution $\mathbf{p}_\theta$.

---

[5]This assumption that the mean is bounded, in our case in $\ell_\infty(\mathbf{0}, 1)$, is standard, and necessary in order to obtain finite upper bounds: indeed, a packing argument shows that if the mean is assumed to be in a ball of radius $R$, then a $\log^{\Omega(1)} R$ dependence in the sample complexity is necessary in both the communication-constrained and LDP settings. Our choice of radius 1 is arbitrary, and our upper bounds can be generalized to any $R \geq 1$.

We now define our measure of interest, the minimax error rate of mean estimation.

*Definition* 3 (Minimax rate of mean estimation). Let $\mathcal{P}$ be a family of distributions parameterized by $\Theta \subseteq \mathbb{R}^d$. For $p \in [1, \infty]$, $n \in \mathbb{N}$, and a family of channels $\mathcal{W}$, the minimax error rate of mean estimation for $\mathcal{P}$ using $\mathcal{W}^n$ under $\ell_p$ loss, denoted $\mathcal{E}_p(\mathcal{P}, \mathcal{W}, n)$, is the least $\gamma \in (0, 1]$ such that there exists an $(n, \gamma)$-estimator for $\mathcal{P}$ using $\mathcal{W}$ under $\ell_p$ loss (see Definition 2).

We obtain lower bounds on the minimax rate of mean estimation for the different families above by specializing our general bound. Importantly, our methodology is not specific to $\ell_p$ losses, and can be used for arbitrary additive losses such as (squared) Hellinger or, indeed, for any loss function for which an analogue of Lemma 1 can be derived.

**Product Bernoulli family.** We first establish the following bounds for $\mathcal{B}_{d,s}$ under privacy and communication constraints.

**Theorem 3** (Product Bernoulli). *Fix $p \in [1, \infty)$. For $4 \log d \le s \le d$, $\varepsilon \in (0, \infty)$, and $\ell \ge 1$,*

$$\sqrt{\frac{ds^{2/p}}{n(\varepsilon^2 \wedge \varepsilon)}} \wedge 1 \lesssim \mathcal{E}_p(\mathcal{B}_{d,s}, \mathcal{W}^{\mathrm{priv},\varepsilon}, n) \lesssim \sqrt{\frac{ds^{2/p}}{n(\varepsilon^2 \wedge 1)}} \tag{10}$$

*and*

$$\sqrt{\frac{ds^{2/p}}{n\ell} \vee \frac{s^{2/p} \log \frac{2d}{s}}{n}} \wedge 1 \lesssim \mathcal{E}_p(\mathcal{B}_{d,s}, \mathcal{W}^{\mathrm{comm},\ell}, n) \lesssim \sqrt{\frac{ds^{2/p}}{n\ell} \vee \frac{s^{2/p} \log \frac{2d}{s}}{n}} \tag{11}$$

*For $p = \infty$, we have the upper bounds*

$$\mathcal{E}_\infty(\mathcal{B}_{d,s}, \mathcal{W}^{\mathrm{priv},\varepsilon}, n) = O\left(\sqrt{\frac{d \log s}{n\varepsilon^2}}\right) \quad and \quad \mathcal{E}_\infty(\mathcal{B}_{d,s}, \mathcal{W}^{\mathrm{comm},\ell}, n) = O\left(\sqrt{\frac{d \log s}{n\ell} \vee \frac{\log d}{n}}\right),$$

*while the lower bounds given in Eqs. (10) and (11) hold for $p = \infty$, too.*[6]

*Remark* 1. Previous work had shown, in the simpler *noninteractive* model, a rate lower bound scaling as $\sqrt{ds/(n\ell) \log(2d/s)}$ for the specific case of $\ell_2$ loss (see, for instance, [21, Theorem 7] for the sparse Gaussian case, which implies the Bernoulli one). An analogous phenomenon was observed for local privacy (*e.g.*, [14]). Thus, by removing this logarithmic factor from the upper bound, our result establishes the first (to the best of our knowledge) separation between interactive and noninteractive protocols for sparse mean estimation under communication or local privacy constraints.

*Remark* 2. Although we stated for simplicity the lower bounds of Theorem 3 in the case where all $n$ players have a similar local constraints (*i.e.*, same privacy parameter $\varepsilon$, or same bandwidth constraint $\ell$), it is immediate to check from the application of Theorem 2 that the result extends to different constraints for each player; replacing $n(\varepsilon^2 \wedge \varepsilon)$ and $n\ell$ in the statement by $\sum_{t=1}^n \varepsilon_t^2 \wedge \varepsilon_t$ and $\sum_{t=1}^n \ell_t$, respectively. A similar remark applies to the Gaussian and discrete families.

**Gaussian family.** We derive a lower bound for $\mathcal{E}_p(\mathcal{G}_{d,s}, \mathcal{W}, n)$ under local privacy (captured by $\mathcal{W} = \mathcal{W}^{\mathrm{priv},\varepsilon}$) and communication (captured by $\mathcal{W} = \mathcal{W}^{\mathrm{comm},\ell}$) constraints.[7] Recall that for product Bernoulli mean estimation we had optimal bounds for both privacy and communication constraints for all finite $p$. For Gaussians, we will obtain tight bounds for privacy constraints for $\varepsilon \in (0, 1]$. However, for communication constraints and privacy constraints when $\varepsilon \ge 1$, our bounds for Gaussian distributions lose a (single) logarithmic factor in some parameter regimes.

**Theorem 4** (Gaussian distributions). *Fix $p \in [1, \infty)$. For $4 \log d \le s \le d$, under LDP constraints, when $\varepsilon \in (0, 1]$,*

$$\sqrt{\frac{ds^{2/p}}{n\varepsilon^2}} \wedge 1 \lesssim \mathcal{E}_p(\mathcal{G}_{d,s}, \mathcal{W}^{\mathrm{priv},\varepsilon}, n) \lesssim \sqrt{\frac{ds^{2/p}}{n\varepsilon^2}} \tag{12}$$

*and when $\varepsilon > 1$,*

$$\sqrt{\frac{ds^{2/p}}{n\varepsilon \log(nd)}} \wedge 1 \lesssim \mathcal{E}_p(\mathcal{G}_{d,s}, \mathcal{W}^{\mathrm{priv},\varepsilon}, n) \lesssim \sqrt{\frac{ds^{2/p}}{n}} \tag{13}$$

---

[6]That is, the upper and lower bounds only differ by a log $s$ factor for $p = \infty$.

[7]As in the Bernoulli case, we here focus for simplicity on the case where the communication (resp., privacy) parameters are the same for all players, but our lower bounds easily extend.

*Under communication constraints,*

$$\sqrt{\frac{ds^{2/p}}{n\ell\log(dn)} \vee \frac{s^{2/p}\log\frac{2d}{s}}{n}} \wedge 1 \lesssim \mathcal{E}_p(\mathcal{G}_{d,s}, \mathcal{W}^{\mathrm{comm},\ell}, n) \lesssim \sqrt{\frac{ds^{2/p}}{n\ell} \vee \frac{s^{2/p}\log\frac{2d}{s}}{n}} \tag{14}$$

*The same bound as in Theorem 3 hold for $p = \infty$.*

We emphasize that, as discussed in Sections 1.1 and 1.2, to the best of our knowledge our results provides the first lower bounds for interactive Gaussian mean estimation under these constraints.

**Discrete distribution estimation.** We are able to derive a lower bound for $\mathcal{E}_p(\Delta_d, \mathcal{W}, n)$, the minimax rate for discrete density estimation, under local privacy and communication constraints. In the interest of space, we focus here on two important corollaries; first, for the case of total variation distance ($\ell_1$), where combining it with known upper bounds we obtain optimal bounds for all $\varepsilon > 0$. In particular, for $\varepsilon \in (0, 1]$ (high-privacy regime) we retrieve the lower bound established in [5], which matches the upper bound from [4]. For $\varepsilon > 1$ (low-privacy regime), our bound matches the upper bound for the noninteractive case, established in [32, 1], showing that even in this low-privacy regime interactivity cannot lead to better rates, except maybe up to constant factors.

**Corollary 3** (Total variation distance)**.** *For $\varepsilon > 0$, we have*

$$\mathcal{E}_1(\Delta_d, \mathcal{W}^{\mathrm{priv},\varepsilon}, n) \asymp \sqrt{\frac{d^2}{n((e^\varepsilon - 1)^2 \wedge e^\varepsilon)}} \wedge 1. \tag{15}$$

*For $\ell \geq 1$,*

$$\mathcal{E}_1(\Delta_d, \mathcal{W}^{\mathrm{comm},\ell}, n) \asymp \sqrt{\frac{d^2}{n(2^\ell \wedge d)}} \wedge 1. \tag{16}$$

For the case of $\ell_2$ estimation, we also obtain order-optimal bounds:

**Corollary 4** ($\ell_2$ density estimation)**.** *For $\varepsilon > 0$, we have*

$$\mathcal{E}_2(\Delta_d, \mathcal{W}^{\mathrm{priv},\varepsilon}, n) \asymp \sqrt{\frac{d}{n(e^\varepsilon - 1)^2 \wedge e^\varepsilon)}} \wedge \sqrt[4]{\frac{1}{n(e^\varepsilon - 1)^2 \wedge e^\varepsilon)}} \wedge 1. \tag{17}$$

*For $\ell \geq 1$,*

$$\mathcal{E}_2(\Delta_d, \mathcal{W}^{\mathrm{comm},\ell}, n) \asymp \sqrt{\frac{d}{n(2^\ell \wedge d)}} \wedge \sqrt[4]{\frac{1}{n(2^\ell \wedge d)}} \wedge 1. \tag{18}$$

# 6 Acknowledgment

The authors would like to thank Yanjun Han, and the anonymous reviewers for helpful comments on an earlier version of this paper.

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

# A  Fully interactive model

In this appendix, we describe how to extend our results, presented in the sequentially interactive model, to the more general interactive setting. We first formally define this setting and the corresponding notion of protocols. Hereafter, we use $^*$ for the Kleene star operation, *i.e.*, $V^* = \bigcup_{n=0}^{\infty} V^n$.

*Definition* 4 (Interactive Protocols). Let $X_1, \ldots, X_n$ be i.i.d. samples from $\mathbf{p}_\theta$, $\theta \in \Theta$, and $\mathcal{W}^*$ be a collection of sequences of pairs of channel families and players; that is, each element of $\mathcal{W}^*$ is a sequence $(\mathcal{W}_t, j_t)_{t \in \mathbb{N}}$ where $j_t \in [n]$. An *interactive protocol* $\Pi$ *using* $\mathcal{W}^*$ comprises a random variable $U$ (independent of the input $X_1, \ldots, X_n$) and, for each $t \in \mathbb{N}$, mappings

$$\sigma_t \colon Y_1, \ldots, Y_{t-1}, U \mapsto N_t \in [n] \cup \{\bot\}$$
$$g_t \colon Y_1, \ldots, Y_{t-1}, U \mapsto W_t$$

with the constraint that $((W_1, N_1), \ldots, (W_t, N_t))$ must be consistent with some sequence from $\mathcal{W}^*$; that is, there exists $((\mathcal{W}_s, j_s))_{s \in \mathbb{N}} \in \mathcal{W}^*$ such that $W_s \in \mathcal{W}_s$ and $N_s = j_s$ for all $1 \le s \le t$. These two mappings respectively indicate (i) whether the protocol is to stop (symbol $\bot$), and, if not, which player is to speak at round $t \in \mathbb{N}$, and (ii)) which channel this player selects at this round.

In round $t$, if $N_t = \bot$, the protocol ends. Otherwise, player $N_t$ (as selected by the protocol, based on the previous messages) uses the channel $W_t$ to produce the message (output) $Y_t$ according to the probability measure $W_t(\cdot \mid X_{N_t})$. We further require that $T := \inf \{ t \in \mathbb{N} : N_t = \bot \}$ is finite a.s. The messages $Y^T = (Y_1, \ldots, Y_T)$ received by the referee and the public randomness $U$ constitute the *transcript* of the protocol $\Pi$.

In other terms, the channel used by the player $N_t$ speaking at time $t$ is a Markov kernel

$$W_t \colon \mathfrak{Y}_t \times \mathcal{X} \times \mathcal{Y}^{t-1} \to [0, 1],$$

with $\mathcal{Y}_t \subseteq \mathcal{Y}$; and, for player $j \in [n]$, the allowed subsequences $(\mathcal{W}_t, j_t)_{t \in \mathbb{N}: j_t = j}$ capture the possible sequences of channels allowed to the player. As an example, if we were to require that any single player can speak at most once, then for every $j \in [n]$ and every $(\mathcal{W}_t, j_t)_{t \in \mathbb{N}} \in \mathcal{W}^n$, we would have $\sum_{t=1}^{\infty} \mathbb{1}\{j_t = j\} \le 1$.

In the interactive model, we can then capture the constraint that each player must communicate at most $\ell$ bits in total by letting $\mathcal{W}^n$ be the set of sequences $(\mathcal{W}_t^{\mathrm{comm}, \ell_t}, j_t)_{t \in \mathbb{N}}$ such that

$$\forall j \in [n], \qquad \sum_{t=1}^{\infty} \ell_t \cdot \mathbb{1}\{j_t = j\} \le \ell.$$

In the simpler sequentially interactive model, this condition simply becomes the choice of $\mathcal{W}^n = (\mathcal{W}^{\mathrm{comm}, \ell}, \ldots, \mathcal{W}^{\mathrm{comm}, \ell})$.

## A.1  Lower Bounds under Full Interactive Model

Next we discuss how our technique extends to the full interactive model. For any full interactive protocol $\Pi$, let $Y^* \in \mathcal{Y}^*$ be the message sequence generated by the protocol. Then, for all $y^* \in \mathcal{Y}^*$, we have

$$\Pr_{X^n \sim \mathbf{p}}[Y^* = y^*] = \mathbb{E}_{X^n \sim \mathbf{p}}\left[\prod_{t=1}^{\infty} W_t\big(y_t \mid X_{\sigma_t(y^{t-1})}, y^{t-1}\big)\right].$$

The following lemma states that if $X^n$ are generated from a product distribution, the distribution of the transcript satisfies a property similar to the "cut-and-paste" property from [6].

**Lemma 2** ([20]). *If $X^n \sim \mathbf{p} = \otimes_{t=1}^n \mathbf{p}_t$, the transcript of the protocol satisfies*

$$\Pr_{X^n \sim \mathbf{p}}[Y^* = y^*] = \prod_{t=1}^{n} \mathbb{E}_{X_t \sim \mathbf{p}_t}[g_t(y^*, X_t)], \tag{19}$$

*where $g_t(y^*, x_t) = \prod_{j=1}^{\infty} W_j(y_j \mid x_t, y^{j-1}) \mathbb{1}\{\sigma_j(y^{j-1}) = t\}$.*

Hence, when $X^n \sim \mathbf{p}_z^{\otimes n}$ we have

$$\mathbf{p}_z^{y^*} := \Pr_{X^n \sim \mathbf{p}_z^{\otimes n}}[Y^* = y^*] = \prod_{t=1}^{n} \mathbb{E}_{X_t \sim \mathbf{p}_z}[g_t(y^*, X_t)].$$

Here we can define a similar notion of "channel" for a communication protocol $\Pi$ for the $i$th player when the underlying distribution is $\mathbf{p}_z$ by setting

$$\tilde{W}_{t,\mathbf{p}_z}(y^* \mid x) = g_t(y^*, x)\left(\prod_{j \neq t} \mathbb{E}_{X_j \sim \mathbf{p}_z}[g_j(y^*, X_j)]\right). \tag{20}$$

Then we have, for all $t \in [n]$,

$$\mathbb{E}_{X_t \sim \mathbf{p}_z}\left[\tilde{W}_{t,\mathbf{p}_z}(y^* \mid X_t)\right] = \Pr_{X^n \sim \mathbf{p}_z^{\otimes n}}[Y^* = y^*].$$

We proceed to prove a bound similar to Theorem 1 in terms of the "channel" defined in Eq. (20), as stated below.

**Theorem 5** (Information contraction bound). *Fix $\tau \in (0, 1/2]$. Let $\Pi$ be a fully interactive protocol using $\mathcal{W}^n$, and let $Z$ be a random variable on $\mathcal{Z}$ with distribution $\mathrm{Rad}(\tau)^{\otimes k}$. Let $(Y^*, U)$ be the transcript of $\Pi$ when the input $X_1, \dots, X_n$ is i.i.d. with common distribution $\mathbf{p}_Z$. Then, under Assumption 1,*

$$\left(\frac{1}{k}\sum_{i=1}^k \mathrm{d}_{\mathrm{TV}}\left(\mathbf{p}_{+i}^{Y^*}, \mathbf{p}_{-i}^{Y^*}\right)\right)^2$$

$$\leq \frac{7}{k}\alpha^2 \sum_{j=1}^n \max_{z \in \mathcal{Z}} \max_{(\mathcal{W}_t, j_t)_{t \in \mathbb{N}} \in \mathcal{W}^n} \sum_{i=1}^k \int_{y^* \in \mathcal{Y}^*} \frac{\mathbb{E}_{\mathbf{p}_z}\left[\phi_{z,i}(X)\tilde{W}_{j,\mathbf{p}_z}(y^* \mid X)\right]^2}{\mathbb{E}_{\mathbf{p}_z}\left[\tilde{W}_{j,\mathbf{p}_z}(y^* \mid X)\right]} \, \mathrm{d}\mu \,,$$

*where $\mathbf{p}_{+i}^{Y^*} := \mathbb{E}\left[\mathbf{p}_Z^{Y^*} \mid Z_i = 1\right]$, $\mathbf{p}_{-i}^{Y^*} := \mathbb{E}\left[\mathbf{p}_Z^{Y^*} \mid Z_i = 1\right]$.*

We can see the bound is in identical form to Theorem 1 except that we replace each player's channel with the $\tilde{W}_{j,\mathbf{p}_z}(y^* \mid X)$ we defined. Other similar bounds in Section 3 can also be derived under additional assumptions and specific constraints. We present the proof for Theorem 5 below and omit the detailed statements and proof for other bounds.

*Proof.* Analogously to Eq. (36), we can get

$$\frac{1}{k}\left(\sum_{i=1}^k \mathrm{d}_{\mathrm{TV}}\left(\mathbf{p}_{+i}^{Y^*}, \mathbf{p}_{-i}^{Y^*}\right)\right)^2 \leq 14 \sum_{t=1}^n \mathbb{E}_Z\left[\sum_{i=1}^k \mathrm{d}_{\mathrm{H}}\left(\mathbf{p}_Z^{Y^*}, \mathbf{p}_{t \leftarrow Z \oplus i}^{Y^*}\right)^2\right] \tag{21}$$

For all $z \in \{-1, +1\}^k$ and $i, t$, by the definition of Hellinger distance and Eq. (19), we have

$$2\mathrm{d}_{\mathrm{H}}\left(\mathbf{p}_z^{Y^*}, \mathbf{p}_{t \leftarrow z \oplus i}^{Y^*}\right)^2 = \int_{y^* \in \mathcal{Y}^*} \prod_{\substack{1 \leq j \leq n \\ j \neq t}} \mathbb{E}_{X_j \sim \mathbf{p}_z}[g_j(y^*, X_j)]\left(\sqrt{\mathbb{E}_{X_t \sim \mathbf{p}_{z \oplus i}}[g_t(y^*, X_t)]} - \sqrt{\mathbb{E}_{X_t \sim \mathbf{p}_z}[g_t(y^*, X_t)]}\right)^2 \mathrm{d}\mu$$

$$\leq \int_{y^* \in \mathcal{Y}^*}\left(\prod_{j \neq t} \mathbb{E}_{X_j \sim \mathbf{p}_z}[g_j(y^*, X_j)]\right)\left(\frac{(\mathbb{E}_{X_t \sim \mathbf{p}_z}[g_t(y^*, X_t)] - \mathbb{E}_{X_t \sim \mathbf{p}_{z \oplus i}}[g_t(y^*, X_t)])^2}{\mathbb{E}_{X_t \sim \mathbf{p}_z}[g_t(y^*, X_t)]}\right) \mathrm{d}\mu \,,$$

Proceeding from above, we get under Assumption 1,

$$2\mathrm{d}_{\mathrm{H}}\left(\mathbf{p}_z^{Y^*}, \mathbf{p}_{t \leftarrow z \oplus i}^{Y^*}\right)^2 \leq \alpha^2 \int_{y^* \in \mathcal{Y}^*}\left(\prod_{j \neq t} \mathbb{E}_{X_j \sim \mathbf{p}_z}[g_j(y^*, X_j)]\right)\left(\frac{\mathbb{E}_{X_t \sim \mathbf{p}_z}[\phi_{z,i}(X_t)g_t(y^*, X_t)]^2}{\mathbb{E}_{X_t \sim \mathbf{p}_z}[g_t(y^*, X_t)]}\right) \mathrm{d}\mu$$

$$= \alpha^2 \int_{y^* \in \mathcal{Y}^*} \frac{\mathbb{E}_{X_t \sim \mathbf{p}_z}\left[\phi_{z,i}(X_t)g_t(y^*, X_t)\prod_{j \neq t}\mathbb{E}_{X_j \sim \mathbf{p}_z}[g_j(y^*, X_j)]\right]^2}{\mathbb{E}_{X_t \sim \mathbf{p}_z}\left[g_t(y^*, X_t)\prod_{j \neq t}\mathbb{E}_{X_j \sim \mathbf{p}_z}[g_j(y^*, X_j)]\right]} \mathrm{d}\mu$$

$$= \alpha^2 \int_{y^* \in \mathcal{Y}^*} \frac{\mathbb{E}_{X_t \sim \mathbf{p}_z}\left[\phi_{z,i}(X_t)\tilde{W}_{t,\mathbf{p}_z}(y^* \mid X)\right]^2}{\mathbb{E}_{X_t \sim \mathbf{p}_z}\left[\tilde{W}_{t,\mathbf{p}_z}(y^* \mid X)\right]} \mathrm{d}\mu \,.$$

Plugging the above bound into Eq. (21), we can obtain the bound in Theorem 5 by taking the maximum over all $z \in \{-1, +1\}^k$ and all possible channel sequences. $\qquad\square$

## B A measure change bound

We here provide a variant of Talagrand's transportation-cost inequality which is used in deriving Eq. (5) (under Assumption 3) in the second part of Theorem 2. We note that this type of result is not novel, and can be derived from standard arguments in the literature (see, e.g., [9, Chapter 8] or [27, Chapter 4]). However, the lemma below is specifically tailored for our purposes, and we provide the proof for completeness. A similar bound was derived in [2], where Gaussian mean testing under communication constraints was considered.

**Lemma 3** (A measure change bound). *Consider a random variable $X$ taking values in $\mathcal{X}$ and with distribution $P$. Let $\Phi\colon \mathcal{X} \to \mathbb{R}^k$ be such that the random vector $\Phi(X)$ is $\sigma^2$-subgaussian. Then, for any function $a\colon \mathcal{X} \to [0,\infty)$ such that $\mathbb{E}[a(X)] < \infty$, we have*

$$\frac{\|\mathbb{E}[\Phi(X)a(X)]\|_2^2}{\mathbb{E}[a(X)]^2} \leq 2\sigma^2 \frac{\mathbb{E}[a(X)\ln a(X)]}{\mathbb{E}[a(X)]} + 2\sigma^2 \ln \frac{1}{\mathbb{E}[a(X)]}.$$

*Proof.* By an application of Gibb's variational principle ($cf.$ [9, Corollary 4.14]) the following holds: For a random variable $Z$ and distributions $P$ and $Q$ on the underlying probability space satisfying $Q \ll P$ (that is, such that $Q$ is absolutely continuous with respect to $P$), we have

$$\lambda \mathbb{E}_Q[Z] \leq \ln \mathbb{E}_P\big[e^{\lambda Z}\big] + \mathrm{D}(Q\|P).$$

To apply this bound, set $P$ to be the distribution of $X$ and let $Q \ll P$ be defined using its density (Radon–Nikodym derivative) with respect to $P$ given by

$$\frac{\mathrm{d}Q}{\mathrm{d}P} = \frac{a(X)}{\mathbb{E}_P[a(X)]}.$$

Now, note that for any unit vector $v$, we have, setting $Z = v^\mathsf{T}\Phi(X)$ and using the $\sigma^2$-subgaussianity of $\Phi(X)$, that

$$\lambda \mathbb{E}_Q[v^\mathsf{T}\Phi(X)] \leq \ln \mathbb{E}_P\Big[e^{\lambda v^\mathsf{T}\Phi(X)}\Big] + \mathrm{D}(Q\|P) \leq \frac{\sigma^2\lambda^2}{2} + \mathrm{D}(Q\|P).$$

In particular, for $\lambda = \frac{1}{\sigma}\sqrt{2\mathrm{D}(Q\|P)}$, we get

$$\mathbb{E}_Q[v^\mathsf{T}\Phi(X)] \leq \sigma\sqrt{2\mathrm{D}(Q\|P)}.$$

Applying this to the unit vector $v := \frac{\mathbb{E}_Q[\Phi(X)]}{\|\mathbb{E}_Q[\Phi(X)]\|_2}$ then yields

$$\|\mathbb{E}_Q[\Phi(X)]\|_2 \leq \sigma\sqrt{2\mathrm{D}(Q\|P)}.$$

To conclude, it then suffices to observe that

$$\mathrm{D}(Q\|P) = \frac{\mathbb{E}_P[a(X)\ln a(X)]}{\mathbb{E}_P[a(X)]} + \ln \frac{1}{\mathbb{E}_P[a(X)]}.$$

The proof is completed by combining the bounds above, as $\mathbb{E}_Q[\Phi(X)] = \frac{\mathbb{E}_P[\Phi(X)a(X))]}{\mathbb{E}_P[a(X)]}$. $\qquad\square$

## C Upper bounds

We now describe and analyze the interactive algorithms for the estimation tasks we consider.

### C.1 Product Bernoulli Distributions

Recall that $\mathcal{B}_{d,s}$, the family of $d$-dimensional $s$-sparse product Bernoulli distributions, is defined as

$$\mathcal{B}_{d,s} := \left\{ \bigotimes_{j=1}^d \mathrm{Rad}(\tfrac{1}{2}(\mu_j + 1)) \ : \ \mu \in [-1,1]^d, \|\mu\|_0 \leq s \right\}. \tag{22}$$

We now provide the interactive protocols achieving the upper bounds of Theorem 3 for sparse product Bernoulli mean estimation under LDP and communication constraints .

Our protocols has two ingredients described below:

- **Estimating non-zero mean coordinates.** In this step we will start with $S_0 = [d]$, the set of all possible coordinates. Then we will iteratively prune the set $S_0 \to S_1 \to \ldots \to S_T$, such that $|S_T| = 3s$ (this step is skipped if $s \geq d/3$) is a good estimate for the set of coordinates with non-zero mean.

- **Estimating the non-zero means.** We then estimate the means of the coordinates in $S_T$, which is equivalent to solving a dense mean estimation problem in $3s$ dimensions.

In the next two sections, we provide the details of the algorithm that matches the lower bounds obtained in Section 5 for interactive protocols under LDP and communication constraints respectively.

### C.1.1 LDP constraints

In this subsection, we will focus on the case $\varepsilon \in (0, 1]$ (high-privacy regime). For the case $\varepsilon > 1$, we rely a privatization of the communication-limited algorithm, which will be discussed at the end of Appendix C.1.2. Our protocol for Bernoulli mean estimation under LDP constraints is described in Algorithm 1. As stated above, in each round $t = 1, \ldots, T$, for each $j \in S_{t-1}$ a new group of players apply the well known binary Randomized Response (RR) mechanism [29, 24] to their $j$th coordinate. Using these messages we then guess a set of coordinates with highest possible means (in absolute value) and prune the set to $S_t$. This is done in Lines 2-6 of Algorithm 1.

In Lines 7-12, the algorithm uses the same approach to estimate the means of coordinates within $S_T$ and sets remaining coordinates to zero.

The privacy guarantee follows immediately from that of the RR mechanism, and further, this only requires one bit of communication per player.

---

**Algorithm 1** LDP protocol for mean estimation for the product of Bernoulli family

---

**Require:** $n$ players, dimension $d$, sparsity parameter $s$, privacy parameter $\varepsilon$.

1: Set $T := \log_3 \frac{d}{3s}$, $\alpha := \frac{e^\varepsilon}{1+e^\varepsilon}$, $S_0 = [d]$, $N_0 := \frac{n}{6d}$.
2: **for** $t = 1, 2, \ldots, T$ **do**
3:      **for** $j \in S_{t-1}$ **do**
4:          Get a group of new players $G_{t,j}$ of size $N_t = N_0 \cdot 2^t$.
5:          Player $i \in G_{t,j}$, upon observing $X_i \in \{-1, +1\}^d$ sends the message $Y_i \in \{-1, +1\}$ such that

$$Y_i = \begin{cases} (X_i)_j & \text{w.p. } \alpha, \\ -(X_i)_j & \text{w.p. } 1 - \alpha. \end{cases} \tag{23}$$

6:          Set $M_{t,j} := \sum_{i \in G_{t,j}} Y_i$. Let $S_t \subseteq S_{t-1}$ be the set of the $|S_{t-1}|/3$ indices with the largest $|M_{t,j}|$.
7: **for** $j \in S_T$ **do**
8:      Get a group of new players $G_{T,j}, j \in S_T$ of size $N_{T+1} = N_0 \cdot 2^T$.
9:      Player $i \in G_{T,j}$, sends the message $Y_i \in \{-1, +1\}$ according to Eq. (23) and $M_{T,j} := \sum_{i \in G_{T,j}} Y_i$
10: **for** $j \in [d]$ **do**
11:

$$\widehat{\mu}_j = \begin{cases} \frac{M_{j,T}}{(2\alpha-1)N_{T+1}} & \text{if } j \in S_T, \\ 0 & \text{otherwise.} \end{cases}$$

12: **return** $\widehat{\mu}$.

---

The performance guarantee of Algorithm 1 is stated below, which matches the lower bounds obtained in Section 5.

**Proposition 1.** *Fix $p \in [1, \infty]$. For $n \geq 1$ and $\varepsilon \in (0, 1]$, Algorithm 1 is an $(n, \gamma)$-estimator using $\mathcal{W}_\varepsilon$ under $\ell_p$ loss for $\mathcal{B}_{d,s}$ with $\gamma = O\left(\sqrt{\frac{pds^{2/p}}{n\varepsilon^2}}\right)$ for $p \leq 2 \log s$ and $\gamma = O\left(\sqrt{\frac{d \log s}{n\varepsilon^2}}\right)$ for $p > 2 \log s$.*

*Proof.* The total number of players used by Algorithm 1 uses is

$$\sum_{t=1}^{T+1} |S_{t-1}| \cdot N_t = |S_0| \cdot N_0 \cdot \sum_{t=1}^{T+1} \frac{2^t}{3^{t-1}} \le 6|S_0| \cdot N_0 = n.$$

To prove the utility guarantee, we bound the estimation error in the estimated set $S_T$ and the error outside the set $S_T$ in the following lemma.

**Lemma 4.** *Let $S_T$ be the subset obtained from the first stage of Algorithm 1. Then,*

$$\max\left\{ \mathbb{E}\left[ \sum_{j \notin S_T} |\mu_j - \widehat{\mu}_j|^p \right], \mathbb{E}\left[ \sum_{j \in S_T} |\mu_j - \widehat{\mu}_j|^p \right] \right\} = O\left( s\left( \frac{pd}{n\varepsilon^2} \right)^{p/2} \right).$$

The proposition follows directly from the lemma. Indeed, for $p > 2 \log s$, by monotonicity of $\ell_p$ norms we have $\|\mu - \hat{\mu}\|_p \le \|\mu - \hat{\mu}\|_{p'}$ for all $p' \le p$, and thus choosing $p' := 2 \log s$ is sufficient to obtain the stated bound. $\square$

*Proof of Lemma 4.* We prove the bound on each term individually. The first term captures the performance of our estimator within coordinates in $S_T$ and the second term states that we do not "prune" too many coordinates with high non-zero means.

**Bounding the first term.** For $j \notin S_T$, we output $\widehat{\mu}_j = 0$. Therefore,

$$\mathbb{E}\left[ \sum_{j \notin S_T} |\mu_j - \widehat{\mu}_j|^p \right] = \sum_j \mathbb{E}[|\mu_j - \widehat{\mu}_j|^p \cdot \mathbb{1}\{j \notin S_T\}] = \sum_j |\mu_j|^p \cdot \Pr[j \notin S_T].$$

Since $\mu$ is $s$-sparse, it will suffice to show that for all $j$ with $|\mu_j| > 0$,

$$|\mu_j|^p \cdot \Pr[j \notin S_T] = O\left( \left( \frac{pd}{n\varepsilon^2} \right)^{p/2} \right). \tag{24}$$

Let

$$H := 20\sqrt{\frac{d}{n(2\alpha - 1)^2}}.$$

Note that for $\varepsilon \in (0, 1]$, we have $2\alpha - 1 \ge \frac{e-1}{e+1}\varepsilon$. Therefore, if $|\mu_j| \le H$, then Eq. (24) holds since $\Pr[j \notin S] \le 1$. We hereafter assume $|\mu_j| > H$, and let $\mu_j = \beta_j H$ with $\beta_j > 1$. Let $E_{t,j}$ be the event that coordinate $j$ is removed in round $t$ given that $j \in S_{t-1}$. Then we have

$$\Pr[j \notin S_T] \le \sum_{t=1}^{T} \Pr[E_{t,j}].$$

We proceed to bound each $\Pr[E_{t,j}]$ separately. Note that for $i \in G_{t,j}$, $Y_i \in \{-1, +1\}$ and by Eq. (23)

$$\mathbb{E}[Y_i] = (2\alpha - 1) \cdot \mu_j = (2\alpha - 1)\beta_j H. \tag{25}$$

Let $a_{t,j}$ be the number of coordinates $j'$ with $\mu_{j'} = 0$ and $|M_{t,j'}| \ge \frac{1}{2}N_t(2\alpha - 1)\beta_j H$. Since we select the $|S_{t-1}|/3$ coordinates with the largest magnitude of the sum, for $j \notin S_t$ to happen at least one of the following must occur: (i) $a_{t,j} > \frac{1}{3}|S_{t-1}| - s$, or (ii) $M_{t,j} < \frac{1}{2}N_t(2\alpha - 1)\beta_j H$.

By Hoeffding's inequality, we have

$$\Pr\left[ M_{t,j} < \frac{1}{2}N_t(2\alpha - 1)\beta_j H \right] \le \exp\left( -\frac{1}{8}N_t((2\alpha - 1)\beta_j H)^2 \right) < \exp\left( -5 \cdot 2^t \beta_j^2 \right).$$

Let $p_{t,j} := e^{-5 \cdot 2^t \beta_j^2}$. Similarly, for any $j'$ such that $\mu_{j'} = 0$,

$$\Pr\left[ |M_{t,j'}| \ge \frac{1}{2}N_t(2\alpha - 1)\beta_j H \right] \le 2p_{t,j}.$$

Since all coordinates are independent, $a_{t,j}$ is binomially distributed with mean at most $2p_{t,j}|S_{t-1}|$. By Markov's inequality, we get

$$\Pr\left[a_{t,j} > \frac{1}{3}|S_{t-1}| - s\right] \leq \frac{\mathbb{E}[a_{t,j}]}{|S_{t-1}|/3 - s} \leq p_{t,j},$$

recalling that $|S_{t-1}| = d3^{t-1} \geq 9s$. By a union bound and summing over $t \in [T]$, we get

$$\Pr[j \notin S_T] \leq \sum_{t=1}^{T} \Pr[E_{t,j}] \leq \sum_{t=1}^{T} 3p_{t,j} = 3\sum_{t=1}^{T} \exp\left(-2^t \cdot 5\beta_j^2\right) \leq 6\exp\left(-5\beta_j^2\right).$$

Not that for $x > 0$, $x^p e^{-x^2} \leq \left(\frac{p}{2e}\right)^{p/2}$. Hence

$$|\mu_j|^p \cdot \Pr[j \notin S_T] \leq 6H^p\beta_j^p e^{-5\beta_j^2} \leq \left(C\frac{pd}{n\varepsilon^2}\right)^{p/2},$$

for some absolute constant $C > 0$, completing the proof.

**Bounding the second term.** Note that $S_T$ is a random variable itself. We show that the bound holds for any realization of $S_T$. We need the following result which follows from standard moment bounds on binomial distributions.

**Fact 1.** *Let $p \geq 1$, $m \in \mathbb{N}$, $0 \leq q \leq 1$, and $N \sim \mathrm{Bin}(m, q)$. Then, $\mathbb{E}[|N - mq|^p] \leq 2^{-p/2}m^{p/2}p^{p/2}$* .

Applying this with $m = N_T \geq \frac{n}{6d}$, the transformation from Bernoulli to $\{-1, +1\}$, and the scaling by $2\alpha - 1$, yields for $j \in S_T$, and using Eq. (25)

$$\mathbb{E}[|\mu_j - \widehat{\mu}_j|^p] \leq \left(\frac{p}{(n/6d)(2\alpha - 1)^2}\right)^{p/2}.$$

Upon summing over $j \in S_T$, we obtain

$$\mathbb{E}\left[\sum_{j \in S_T} |\mu_j - \widehat{\mu}_j|^p\right] \leq 3s \cdot \left(\frac{6(e+1)^2 d}{(e-1)^2 n\varepsilon^2}\right)^{p/2} \leq 3 \cdot 6^p \cdot s\left(\frac{pd}{n\varepsilon^2}\right)^{p/2}. \qquad \square$$

### C.1.2 Communication constraints

In Algorithm 2 we propose a protocol to estimate the mean of product Bernoulli distributions under $\ell$-bit communication constraints. As mentioned in the previous subsection, the $\varepsilon$-LDP algorithm with $\varepsilon > 1$ will follow from a simple modification of the communication-constrained one; we discuss how to privatize the latter to obtain the former at the end of the section. As in the LDP case when $\varepsilon \in (0, 1]$, in 2–10 the algorithm iteratively prunes an initial set $S_0 = [d]$ to obtain a set $S_T$ of size $\max\{3s, \ell\}$, which denotes the set of potential non-zero coordinates. We then estimate the mean of coordinates in $S_T$. If $\ell > 3s$, then we can directly send the values of all coordinates in $S_T$ and use it for estimation; otherwise, when $3s > \ell$, we again partition $S_T$ into sets of size $\ell$ and each player sends the bits of its sample in this set. This is done in Lines 11–18. We state the performance of Algorithm 2 below.

**Proposition 2.** *Fix $p \in [1, \infty]$. For $n \geq 1$ and $\ell \leq d$, we have Algorithm 2 is an $(n, \gamma)$-estimator using $\mathcal{W}_\ell$ under $\ell_p$ loss for $\mathcal{B}_{d,s}$ with $\gamma = O\left(\sqrt{\frac{pds^{2/p}}{n\ell} + \frac{(p+\log(2\ell/s))s^{2/p}}{n}}\right)$ for $p \leq 2\log s$ and*

*$\gamma = O\left(\sqrt{\frac{d\log s}{n\ell} + \frac{\log \ell}{n}}\right)$ for $p > 2\log s$.*

When $\ell \leq 3s$, the bound we get is $\gamma \lesssim \sqrt{\frac{pds^{2/p}}{n\ell}}$. The analysis is almost identical to the case under LDP constraints, since in both cases, the information we get about coordinate $j$ are samples from a Rademacher distribution with mean $(2\alpha - 1)\mu_j$. There are only two differences. (i) $\alpha = 1$ instead of

**Algorithm 2** $\ell$-bit protocol for estimating product of Bernoulli family

---

**Require:** $n$ players, dimension $d$, sparsity parameter $s$, communication bound $\ell$.

1: Set $T := \log_3(d/\max\{3s, \ell\})$, $S_0 := [d]$, $N_0 := \frac{n\ell}{18d}$.
2: **for** $t = 1, 2, \ldots, T$ **do**
3:     Set $P := \frac{d}{3^{t-1}\ell}$, and partition $S_{t-1}$ into $P$ subsets $S_{t-1,1}, \ldots, S_{t-1,P}$, each of size $\ell$.
4:     **for** $j = 1, 2, \ldots, P$ **do**
5:         Get a group of new players $G_{t,j}$ of size $N_t = N_0 \cdot 2^t$.
6:         Player $i \in G_{t,j}$, upon observing $X_i \in \{-1, +1\}^d$ sends the message $Y_i = \{(X_i)_x\}_{x \in S_{t-1,j}}$.
7:         For $x \in S_{t-1,j}$, let $M_{t,x} := \sum_{i \in G_{t,j}} (X_i)_x$.
8:     Set $S_t \subseteq S_{t-1}$ to be the set of indices with the largest $|M_{t,x}|$ and $|S_t| = |S_{t-1}|/3$.
9: **if** $\ell \leq 3s$ **then**
10:     Partition $S_T$ into $3s/\ell$ subsets of size $\ell$ each, $S_{T,j}, j \in [3s/\ell]$.
11:     **for** $j = 1, \ldots, 3s/\ell$ **do**
12:         Get a new group $G_{T+1,j}$ of players of size $n\ell/(6s)$.
13:         Player $i \in G_{T+1,j}$, sends the message $Y_i = \{(X_i)_x\}_{x \in S_{T,j}}$.
14:         For $x \in S_{T,j}$, let $M_{T+1,x} = \sum_{i \in G_{T+1,j}} (X_i)_x$. Set

$$\widehat{\mu}_x := \frac{6s}{n\ell} M_{T+1,x},$$

15:         For $x \notin S_T$, set $\widehat{\mu}_x = 0$.
16: **if** $\ell > 3s$ **then**,
17:     Get $n/2$ new players $G_{T+1}$ and for $i \in G_{T+1}$, player $i$ sends $Y_i = \{(X_i)_x\}_{x \in S_T}$ ▷ This can be done since $|S_T| = \ell$ if $\ell > 3s$.
18:     For $x \in S_T$, let $M_{T+1,x} = \sum_{i \in G_{T+1,j}} (X_i)_x$. Set $S_{T+1} \subseteq S_T$ to be the set of indices with the largest $|M_{T+1,x}|$ and $|S_{T+1}| = 3s$. For all $x \in S_{T+1}$, set

$$\widehat{\mu}_x := \frac{2}{n} M_{T+1,x},$$

and for all $x \notin S_{T+1}, \widehat{\mu}_x = 0$.
19: **return** $\widehat{\mu}$.

---

$\Theta(\varepsilon^2)$. (ii) There is a factor of $\ell$ more players in the corresponding groups. Combing both factors, we can obtain the desired bound by replacing $\varepsilon^2$ by $\ell$. We omit the detailed proof in this case.

When $\ell > 3s$, after $T \asymp \log(d/\ell)$ rounds, we can find a subset $S_T$ of size $\ell$ which contains most of the coordinates with large biases. The protocol then asks new players to send all coordinates within $S_T$ using $\ell$ bits. In this case, it would be enough to prove Lemma 5 since for the coordinates outside $S_T$, we can show the error is small following exactly the same steps as the proof for bouding the first term in Lemma 4 as we explained in the case when $\ell \leq 3s$.

**Lemma 5.** *Let $S_T$ be the subset obtained from the first stage of Algorithm 2, we have*

$$\mathbb{E}\left[\sum_{j \in S_T} |\mu_j - \widehat{\mu}_j|^p\right] = O\left(s\left(\frac{p + \log \frac{2\ell}{s}}{n}\right)^{p/2}\right).$$

*Proof.* Similar to Lemma 4, we will prove that the statement is true for any realization of $S_T$, which is a stronger statement than the claim.

$$\mathbb{E}\left[\sum_{j \in S_T} |\mu_j - \widehat{\mu}_j|^p\right] = \mathbb{E}\left[\sum_{j \in S_T} |\mu_j - \widehat{\mu}_j|^p \mathbb{1}\{j \in S_{T+1}\}\right] + \mathbb{E}\left[\sum_{j \in S_T} |\mu_j|^p \mathbb{1}\{j \notin S_{T+1}\}\right]$$

$$\leq \mathbb{E}\left[\sum_{j \in S_{T+1}} |\mu_j - \widehat{\mu}_j|^p\right] + \sum_{j \in S_T} |\mu_j|^p \Pr[j \notin S_{T+1}].$$

Fix $S_{T+1}$. For each $j \in S_{T+1}$, $M_{T+1,j}$ is binomially distributed with mean $\mu_j$ and $n/2$ trials. By similar computations as Lemma 4, we have

$$\mathbb{E}\left[\sum_{j \in S_{T+1}} |\mu_j - \widehat{\mu}_j|^p\right] = O\left(s\left(\frac{p}{n}\right)^{p/2}\right). \tag{26}$$

Next we show for all $j \in S_T$ such that $\mu_j \neq 0$,

$$|\mu_j|^p \Pr[j \notin S_{T+1}] \leq 2\left(\frac{p \vee 64 \ln \frac{2\ell}{s}}{n}\right)^{p/2}. \tag{27}$$

If $|\mu_j| \leq H' := 8\sqrt{\frac{\ln \frac{2\ell}{s}}{n}}$, Eq. (27) always holds since $\Pr[j \notin S] \leq 1$. Hence we hereafter assume that $|\mu_j| > H'$, and write $\mu_j = \beta_j H'$ for some $\beta_j > 1$.

Let $a_{T+1,j}$ be the number of coordinates $j'$ with $\mu_{j'} = 0$ and $|M_{T+1,j'}| \geq \frac{n}{2} \cdot \frac{\beta_j H'}{2}$. Then since $S_{T+1}$ contains the top $3s$ coordinates with the largest magnitude of the sum, we have $j \notin S_{T+1}$ happens only if at least one of the following occurs (i) $a_{T+1,j} > 2s$, or (ii) $M_{T+1,j} < \frac{n}{2} \cdot \frac{\beta_j H'}{2}$.

By Hoeffding's inequality, we have

$$\Pr\left[M_{T+1,j} < \frac{n}{2} \cdot \frac{\beta_j H'}{2}\right] \leq \exp\left(-\frac{1}{2} \cdot \frac{n}{2} \cdot \left(\frac{\beta_j H'}{2}\right)^2\right) = \left(\frac{2\ell}{s}\right)^{-4\beta_j^2} := p_{T+1,j}.$$

Similarly, for any $j'$ such that $\mu_{j'} = 0$,

$$\Pr\left[|M_{T+1,j'}| \geq \frac{n}{2} \cdot \frac{\beta_j H'}{2}\right] \leq 2p_{T+1,j}.$$

Since all coordinates are independent, $a_{T+1,j}$ is binomially distributed with mean at most $2p_{T+1,j}\ell$, and therefore, by Markov's inequality,

$$\Pr[a_{T+1,j} > 2s] \leq \frac{2p_{T+1,j}\ell}{2s} \leq \left(\frac{2\ell}{s}\right)^{1-4\beta_j^2} \leq \left(\frac{2\ell}{s}\right)^{-3\beta_j^2}$$

the last step since $\beta_j > 1$. By a union bound, we have

$$\Pr[j \notin S_T] \leq \Pr[a_{T+1,j} > 2s] + \Pr\left[M_{T+1,j} < \frac{1}{4}\frac{n}{2} \cdot \frac{\beta_j H'}{2}\right] \leq 2\left(\frac{2\ell}{s}\right)^{-3\beta_j^2}.$$

Using the inequality $x^p a^{-x^2} \leq \left(\frac{p}{2e \ln a}\right)^{p/2}$ which holds for all $x > 0$, we get overall

$$|\mu_j|^p \cdot \Pr[j \notin S_T] \leq 2H'^p \beta_j^p \left(\frac{2\ell}{s}\right)^{-4\beta_j^2} \leq 2\left(\frac{p}{en}\right)^{p/2},$$

establishing Eq. (27). Combining Eq. (26) and Eq. (27) concludes the proof Lemma 5 since there are at most $s$ unbiased coordinates. $\square$

**Algorithm under LDP with $\varepsilon > 1$**   To get a $\varepsilon$-LDP algorithm in the regime $\varepsilon > 1$ (low-privacy regime), we perform the following changes to obtain a private algorithm from Algorithm 2:

- Each user independently flips each coordinate of their local sample to get $Z_i$ where, for all $x \in [d]$, $(Z_i)_x = (X_i)_x$ with probability $\frac{e}{e+1}$ and $(Z_i)_x = 1 - (X_i)_x$ with probability $\frac{1}{e+1}$ (note that this corresponds to applying Randomized Response independently to each bit with privacy parameter 1).
- Users then follow Algorithm 2 with the setting $\ell = \lfloor \varepsilon \rfloor$ and local data $\{Z_i\}_{i \in [n]}$, and obtain estimate $\widehat{\mu}$.
- The final estimate is then $\frac{e+1}{e-1}\widehat{\mu}$.

The privacy guarantee of the algorithm comes from the fact that Algorithm 2 sends at most $\ell = \lfloor \varepsilon \rfloor$ coordinates of each $Z_i$, and for any $S$ with $|S| \leq \lfloor \varepsilon \rfloor$

$$\frac{\Pr[\{(Z_i)_x\}_{x \in S} \mid X_i]}{\Pr[\{(Z_i)_x\}_{x \in S} \mid X_i']} = \prod_{x \in S} \frac{\Pr[(Z_i)_x \mid (X_i)_x]}{\Pr[(Z_i)_x \mid (X_i')_x]} \leq e^{\lfloor \varepsilon \rfloor}.$$

The utility guarantee follows from observing that $\mu_Z = \frac{e-1}{e+1}\mu$ and hence any $\ell_p$ error guarantee will be preserved up to a constant.

## C.2 Gaussian Mean Estimation

Recall that $\mathcal{G}_{d,s}$ denotes the family of $d$-dimensional spherical Gaussian distributions with $s$-sparse mean in $[-1,1]^d$, *i.e.*,

$$\mathcal{G}_{d,s} = \{\, \mathcal{G}(\mu, \mathbb{I}) \,:\, \|\mu\|_\infty \leq 1, \|\mu\|_0 \leq s \,\}. \tag{28}$$

We will prove the following results for LDP and communication constraints, respectively.

**Proposition 3.** *Fix $p \in [1, \infty]$. For $n \geq 1$ and $\varepsilon \in (0, 1]$, there exists an $(n, \gamma)$-estimator using $\mathcal{W}_\varepsilon$ under $\ell_p$ loss for $\mathcal{G}_{d,s}$ with $\gamma = O\left(\sqrt{\frac{pds^{2/p}}{n\varepsilon^2}}\right)$ for $p \leq 2\log s$ and $\gamma = O\left(\sqrt{\frac{d\log s}{n\varepsilon^2}}\right)$ for $p > 2\log s$.*

**Proposition 4.** *Fix $p \in [1, \infty]$. For $n \geq 1$ and $\ell \leq d$, there exists an $(n, \gamma)$-estimator using $\mathcal{W}_\ell$ under $\ell_p$ loss for $\mathcal{G}_{d,s}$ with $\gamma = O\left(\sqrt{\frac{pds^{2/p}}{n\ell} + \frac{(p+\log(2\ell/s))s^{2/p}}{n}}\right)$ for $p \leq 2\log s$ and $\gamma = O\left(\sqrt{\frac{d\log s}{n\ell} + \frac{\log \ell}{n}}\right)$ for $p > 2\log s$.*

We reduce the problem of Gaussian mean estimation to that of Bernoulli mean estimation and then invoke Propositions 1 and 2 from the previous section. At the heart of the reduction is a simple idea that was used in, *e.g.*, [10, 2, 11]: the sign of a Gaussian random variable already preserves sufficient information about the mean. Details follow.

Let $\mathbf{p} \in \mathcal{G}_{d,s}$ with mean $\mu(\mathbf{p}) = (\mu(\mathbf{p})_1, \ldots, \mu(\mathbf{p})_d)$. For $X \sim \mathbf{p}$, let $Y = (\text{sign}(X_i))_{i \in [d]} \in \{-1, +1\}^d$ be a random variable indicating the signs of the $d$ coordinates of $X$. By the independence of the coordinates of $X$, note that $Y$ is distributed as a product Bernoulli distribution (in $\mathcal{B}_d$) with mean vector $\nu(\mathbf{p})$ given by

$$\nu(\mathbf{p})_i = 2 \Pr_{X \sim \mathbf{p}}[X_i > 0] - 1 = \text{Erf}\left(\frac{\mu(\mathbf{p})_i}{\sqrt{2}}\right), \qquad i \in [d], \tag{29}$$

and, since $|\mu(\mathbf{p})_i| \leq 1$, we have $\nu(\mathbf{p}) \in [-\eta, \eta]^d$, where $\eta := \text{Erf}(1/\sqrt{2}) \approx 0.623$. Moreover, it is immediate to see that each player, given a sample from $\mathbf{p}$, can convert it to a sample from the corresponding product Bernoulli distribution. We now show that a good estimate for $\nu(\mathbf{p})$ yields a good estimate for $\mu(\mathbf{p})$.

**Lemma 6.** *Fix any $p \in [1, \infty)$, and $\mathbf{p} \in \mathcal{G}_d$. For $\widehat{\nu} \in [-\eta, \eta]^d$, define $\widehat{\mu} \in [-1, 1]^d$ by $\widehat{\mu}_i := \sqrt{2}\,\text{Erf}^{-1}(\widehat{\nu}_i)$, for all $i \in [d]$. Then*

$$\|\mu(\mathbf{p}) - \widehat{\mu}\|_p \leq \sqrt{\frac{e\pi}{2}} \cdot \|\nu(\mathbf{p}) - \widehat{\nu}\|_p.$$

*Proof.* By computing the maximum of its derivative,[8] we observe that the function $\text{Erf}^{-1}$ is $\frac{\sqrt{e\pi}}{2}$-Lipschitz on $[-\eta, \eta]$. By the definition of $\widehat{\mu}$ and recalling Eq. (29), we then have

$$\|\mu(\mathbf{p}) - \widehat{\mu}\|_p^p = \sum_{i=1}^d |\mu(\mathbf{p})_i - \widehat{\mu}_i|^p = 2^{p/2} \cdot \sum_{i=1}^d \left|\text{Erf}^{-1}(\nu_i) - \text{Erf}^{-1}(\widehat{\nu}_i)\right|^p \leq \left(\frac{e\pi}{2}\right)^{p/2} \cdot \sum_{i=1}^d |\nu_i - \widehat{\nu}_i|^p,$$

where we used the fact that $\nu, \widehat{\nu} \in [-\eta, \eta]^d$. □

---

[8] Specifically, we have that $\max_{x \in [-\eta, \eta]} \text{Erf}^{-1}(x) = 1/\sqrt{2}$ by definition of $\eta$ and monotonicity of Erf. Recalling then that, for all $x \in [-\eta, \eta]$, $(\text{Erf}^{-1})'(x) = \frac{1}{\text{Erf}'(\text{Erf}^{-1}(x))} = \frac{\sqrt{\pi}}{2}e^{(\text{Erf}^{-1}(x))^2} \leq \frac{\sqrt{\pi}}{2}e^{\frac{1}{2}}$, we get the Lipschitzness claim.

As previously discussed, combining Lemma 6 with Propositions 1 and 2 (with $\gamma' := \sqrt{\frac{2}{e\pi}}\gamma$) immediately implies Propositions 3 and 4 for $p \in [1, \infty]$.

*Remark* 3. Note that for the Gaussian family, we also consider the linear measurement constraint. Under linear measurement constraints, we can use the linear measurement matrix to obtain $r$ out of $d$ coordinates and perform the above reduction to product of Bernoulli family. The obtained bound will be same as that under communication constraints.

# D    Relation to other lower bound methods

We now discuss how our techniques compare with other existing approaches for proving lower bounds under information constraints. Specifically, we clarify the relationship between our technique and the approach using strong data processing inequalities (SDPI) as well as that based on van Trees inequality (a generalization of the Cramér–Rao bound).

## D.1    Strong data processing inequalities

We note first that the bound in Eq. (5) can be interpreted as a strong data processing inequality. Indeed, the average discrepancy on the left-side of inequality can be viewed as the average information $Y^n$ reveals about each bit of $Z$. Here the information is measured in terms of total variation distance. The information quantity on the right-side denotes the information between the input $X^n$ and the output $Y^n$ of the channels. Since the Markov relation $Z^n$ — $X^n$ — $Y^n$ holds, the inequality is thus a strong data processing inequality with strong data processing constant roughly $\sigma^2/k$. Such strong data processing inequalities were used to derive lower bounds for statistical estimation under communication constraints in [34, 10, 31]. We note that our approach recovers these bounds, and further applies to arbitrary constraints captured by $\mathcal{W}$.

## D.2    Connection to the van Trees inequality

The average information bound in (3), in fact, allows us to recover bounds similar to the van Trees inequality-based bounds developed in [7] and [8].

For $\Theta \subset \mathbb{R}^k$ and a parametric family[9] $\mathcal{P}_\Theta = \{\mathbf{p}_\theta, \theta \in \Theta\}$, recall that the Fisher information matrix $J(\theta)$ is a $k \times k$ matrix given by, under some mild regularity conditions,

$$J(\theta)_{i,j} = -\mathbb{E}_{\mathbf{p}_\theta}\left[\frac{\partial^2 \log \mathbf{p}_\theta}{\partial \theta_i \partial \theta_j}(X)\right], \quad i, j \in [k].$$

In particular, the diagonal entries equal

$$J(\theta)_{i,i} = \mathbb{E}_{\mathbf{p}_\theta}\left[\left(\frac{1}{\mathbf{p}_\theta(X)} \cdot \frac{\partial \mathbf{p}_\theta}{\partial \theta_i}(X)\right)^2\right], \quad i \in [k].$$

For our application, given a channel $W \in \mathcal{W}$, we consider the family $\mathcal{P}_\Theta^W := \{\mathbf{p}_\theta^W, \theta \in \Theta\}$ of distributions induced on the output of the channel $W$ when the input distributions are from $\mathcal{P}_\Theta$. We denote the Fisher information matrix for this family by $J^W(\theta)$, which we compute next under a refined version of our Assumption 1 described below.

Let $\theta$ be a point in the interior of $\Theta$ and $\mathbf{p}_\theta$ be differentiable at $\theta$. We set $\theta_z := \theta + \frac{\gamma}{2}z$, $z \in \{-1, +1\}^k$, and make the following assumption about the structure of the parametric family of distribution: For all $z \in \{-1, +1\}^k$ and $i \in [k]$,

$$\frac{\mathrm{d}\mathbf{p}_{z \oplus i}}{\mathrm{d}\mathbf{p}_z} = 1 + \gamma \xi_{z,i}^\gamma + \gamma^2 \psi_{z,i}^\gamma, \tag{30}$$

where $\mathbb{E}_{\mathbf{p}_z}\left[\xi_{z,i}^\gamma(X)^2\right]$ and $\mathbb{E}_{\mathbf{p}_z}\left[\psi_{z,i}^\gamma(X)^2\right]$ are assumed to be uniformly bounded for $\gamma$ sufficiently small; for concreteness, we assume $\mathbb{E}_{\mathbf{p}_z}\left[\psi_{z,i}^\gamma(X)^2\right] \leq c^2$ for a constant $c$, for all $\gamma$ sufficiently small. Let $\xi_{z,i}(x) := \lim_{\gamma \to \mathbf{0}} \xi_{z,i}^\gamma(x)$, for all $x$.

---

[9]We assume that each distribution $\mathbf{p}_\theta$ has a density with respect to a common measure $\nu$, and, with a slight abuse of notation, denote the density of $\mathbf{p}_\theta$ also by $\mathbf{p}_\theta(X)$.

In applications, we expect the dependence of $\xi_{z,i}^{\gamma}$ on $\gamma$ to be "mild," and, in essence, the assumption above provides a linear expansion of the term $\alpha_{z,i}\phi_{z,i}$ from Assumption 1 as a function of the perturbation parameter $\gamma$. Assuming that the densities are differentiable as a function of $\theta$, for the distribution $\mathbf{p}_{\theta}^{W}$ of the output of a channel $W$ with input $X \sim \mathbf{p}_{\theta}$, we get

$$
\begin{aligned}
\frac{\partial \mathbf{p}_{\theta}^{W}(y)}{\partial \theta_i} &= z_i \lim_{\gamma \to 0} \frac{\mathbf{p}_{\theta_z}^{W}(y) - \mathbf{p}_{\theta_z \oplus i}^{W}(y)}{\gamma} \\
&= z_i \lim_{\gamma \to 0} \mathbb{E}_{\mathbf{p}_z}\left[(\xi_{z,i}^{\gamma}(X) + \gamma \psi_{z,i}^{\gamma}(X))W(y \mid X)\right] \\
&= z_i \mathbb{E}_{\mathbf{p}_\theta}[\xi_{z,i}W(y \mid X)],
\end{aligned}
$$

where we used Eq. (30), the fact that $\lim_{\gamma \to 0} \theta_z = \theta$, the fact that $\mathbb{E}_{\mathbf{p}_z}\left[\psi_{z,i}^{\gamma}(X)W(y \mid X)\right] \leq c\sqrt{\mathbb{E}_{\mathbf{p}_z}[W(y \mid X)^2]} \leq c$, and the dominated convergence theorem. Thus, we get

$$
\mathrm{Tr}\big(J^{W}(\theta)\big) = \sum_{i=1}^{k} \int_{\mathcal{Y}} \frac{\mathbb{E}_{\mathbf{p}_\theta}[\xi_{z,i}(X)W(y \mid X)]^2}{\mathbb{E}_{\mathbf{p}_\theta}[W(y \mid X)]} \, \mathrm{d}\mu \,. \tag{31}
$$

Our information contraction bound will be seen later (Section 5) to yield lower bounds for expected estimation error. For concreteness, we give a preview of a version here. We assume for simplicity that $\mathcal{W}_t = \mathcal{W}$ for all $t$ and consider the $\ell_2$ loss function for the dense ($\tau = 1/2$) case. By following the proof of Lemma 1 below, given an $(n,\gamma)$-estimator $\hat{\theta} = \hat{\theta}(Y^n, U)$ of $\mathcal{P}_{\Theta}$ using $\mathcal{W}^n$ under $\ell_2$ loss, we can find an estimator $\hat{Z} = \hat{Z}(Y^n, U)$ such that

$$
\gamma^2 \sum_{i=1}^{k} \Pr\left[\hat{Z}_i \neq Z_i\right] = \mathbb{E}\left[\left\|\theta_Z - \theta_{\hat{Z}}\right\|_2^2\right] \leq 4\gamma^2,
$$

whereby

$$
\frac{1}{k}\sum_{i=1}^{k} \mathrm{d}_{\mathrm{TV}}\left(\mathbf{p}_{+i}^{Y^n}, \mathbf{p}_{-i}^{Y^n}\right) \geq 1 - \frac{2}{k}\sum_{i=1}^{k}\Pr\left[\hat{Z}_i \neq Z_i\right] \geq 1 - \frac{8\gamma^2}{k\gamma^2}\,.
$$

Upon setting $\gamma := 4\gamma/\sqrt{k}$, we get that the left-side of Eq. (3) is bounded below by $1/4$. For the same $\gamma$ and under Eq. (30), the right-side evaluates to

$$
\begin{aligned}
&\frac{4\gamma^2 n}{k} \max_{z \in \mathcal{Z}} \max_{W \in \mathcal{W}} \sum_{i=1}^{k} \int_{\mathcal{Y}} \frac{\mathbb{E}_{\mathbf{p}_z}\left[(\xi_{z,i}^{\gamma}(X) + \gamma \psi_{z,i}^{\gamma}(X))W(y \mid X)\right]^2}{\mathbb{E}_{\mathbf{p}_z}[W(y \mid X)]} \, \mathrm{d}\mu \\
&\leq \frac{8\gamma^2 n}{k} \max_{z \in \mathcal{Z}} \max_{W \in \mathcal{W}} \sum_{i=1}^{k} \int_{\mathcal{Y}} \frac{\mathbb{E}_{\mathbf{p}_z}\left[\xi_{z,i}^{\gamma}(X)W(y \mid X)\right]^2 + \gamma^2 \mathbb{E}_{\mathbf{p}_z}\left[\psi_{z,i}^{\gamma}(X)W(y \mid X)\right]^2}{\mathbb{E}_{\mathbf{p}_z}[W(y \mid X)]} \, \mathrm{d}\mu \\
&\leq \frac{128\gamma^2 n}{k^2} \left( \max_{z \in \mathcal{Z}} \max_{W \in \mathcal{W}} \sum_{i=1}^{k} \int_{\mathcal{Y}} \frac{\mathbb{E}_{\mathbf{p}_z}\left[\xi_{z,i}^{\gamma}(X)W(y \mid X)\right]^2}{\mathbb{E}_{\mathbf{p}_z}[W(y \mid X)]} \, \mathrm{d}\mu + c^2\gamma^2 \right),
\end{aligned}
$$

where we used $(a+b)^2 \leq 2(a^2+b^2)$ and

$$
\int_{\mathcal{Y}} \frac{\mathbb{E}_{\mathbf{p}_z}\left[\psi_{z,i}^{\gamma}(X)W(y \mid X)\right]^2}{\mathbb{E}_{\mathbf{p}_z}[W(y \mid X)]} \, \mathrm{d}\mu \leq \int_{\mathcal{Y}} \mathbb{E}_{\mathbf{p}_z}\left[\psi_{z,i}^{\gamma}(X)^2 W(y \mid X)\right] \mathrm{d}\mu = \mathbb{E}_{\mathbf{p}_z}\left[\psi_{z,i}^{\gamma}(X)^2\right] \leq c^2 \,.
$$

Therefore, Eq. (3) yields

$$
\gamma^2 \geq \frac{k^2}{256 \cdot n \left( \max_{z \in \mathcal{Z}} \max_{W \in \mathcal{W}} \sum_{i=1}^{k} \int_{\mathcal{Y}} \frac{\mathbb{E}_{\mathbf{p}_z}\left[\xi_{z,i}^{\gamma}(X)W(y|X)\right]^2}{\mathbb{E}_{\mathbf{p}_z}[W(y|X)]} \, \mathrm{d}\mu + c^2 \right)} \,.
$$

This bound is, in effect, the same as the van Trees inequality with $\mathrm{Tr}\big(J^{W}(\theta)\big)$ replaced by

$$
g(\gamma) := \sum_{i=1}^{k} \int_{\mathcal{Y}} \frac{\mathbb{E}_{\mathbf{p}_z}[\phi_{z,i}(X)W(y \mid X)]^2}{\mathbb{E}_{\mathbf{p}_z}[W(y \mid X)]} \, \mathrm{d}\mu \,.
$$

In fact, in view of Eq. (31), $\mathrm{Tr}\big(J^W(\theta)\big) = \lim_{\gamma \to 0} g(\gamma) =: g(0)$. Thus, our general lower bound will recover van Trees inequality-based bounds when Eq. (30) holds and $g(\gamma) \approx g(0)$. We note that Eq. (30) holds for all the families considered in this paper (see Eq. (40) for product Bernoulli, Eq. (42) for Gaussian, and Eq. (50) for discrete distributions). We close this discussion by noting that results in Section 3 are obtained by deriving bounds for $g(\gamma)$ which apply for all $\gamma$ and, therefore, also for $g(0) = \mathrm{Tr}\big(J^W(\theta)\big)$.

# E    Missing proofs in Section 3

## E.1    Proof of Theorem 1

Consider $Z = (Z_1, \ldots, Z_k) \in \{-1, 1\}^k$ where $Z_1, \ldots, Z_k$ are i.i.d. with $\Pr[Z_i = 1] = \tau$. For a fixed $i \in [k]$, let

$$\mathbf{p}_{+i}^{Y^n} := \mathbb{E}_Z\Big[\mathbf{p}_Z^{Y^n} \mid Z_i = +1\Big] = \sum_{z: z_i = +1} \Big(\prod_{j \neq i} \tau^{\frac{1+z_j}{2}}(1-\tau)^{\frac{1-z_j}{2}}\Big)\mathbf{p}_z^{Y^n}$$

$$\mathbf{p}_{-i}^{Y^n} := \mathbb{E}_Z\Big[\mathbf{p}_Z^{Y^n} \mid Z_i = -1\Big] = \sum_{z: z_i = -1} \Big(\prod_{j \neq i} \tau^{\frac{1+z_j}{2}}(1-\tau)^{\frac{1-z_j}{2}}\Big)\mathbf{p}_z^{Y^n},$$

the partial mixtures of message distributions conditioned on $Z_i$. We will rely on the following lemma, which relates the desired average discrepancy between the $\mathbf{p}_{+i}^{Y^n}$ and $\mathbf{p}_{-i}^{Y^n}$'s to the sum of $n$ "local" discrepancy measures (in the form of Hellinger distances between local messages). Each local measure can then be easily bounded in terms of the density $\mathbf{p}_z$ and the channel $W$ to get the desired bound.

**Lemma 7.** *With the notation of Theorem 1, we have*

$$\Big(\frac{1}{k}\sum_{i=1}^k \mathrm{d}_{\mathrm{TV}}\Big(\mathbf{p}_{+i}^{Y^n}, \mathbf{p}_{-i}^{Y^n}\Big)\Big)^2 \leq \frac{14}{k}\sum_{t=1}^n \max_{z \in \mathcal{Z}} \max_{W \in \mathcal{W}_t} \sum_{i=1}^k \mathrm{d}_{\mathrm{H}}\big(\mathbf{p}_z^W, \mathbf{p}_{z \oplus i}^W\big)^2, \tag{32}$$

*where $\mathbf{p}_z^W$ denotes the distribution of $Y \sim W(\cdot \mid X)$ when $X \sim \mathbf{p}_z$.*

The proof of the lemma is rather involved and constitutes the core of the argument. We defer it to the end of the section and show first how it implies Theorem 1. For all $z$ and $W$, we have

$$\mathrm{d}_{\mathrm{H}}\big(\mathbf{p}_z^W, \mathbf{p}_{z \oplus i}^W\big)^2 = \frac{1}{2}\int_{y \in \mathcal{Y}} \Big(\sqrt{\mathbb{E}_{\mathbf{p}_z}[W(y \mid X)]} - \sqrt{\mathbb{E}_{\mathbf{p}_{z \oplus i}}[W(y \mid X)]}\Big)^2 \mathrm{d}\mu$$

$$= \frac{1}{2}\int_{\mathcal{Y}} \Big(\frac{\mathbb{E}_{\mathbf{p}_z}[W(y \mid X)] - \mathbb{E}_{\mathbf{p}_{z \oplus i}}[W(y \mid X)]}{\sqrt{\mathbb{E}_{\mathbf{p}_z}[W(y \mid X)]} + \sqrt{\mathbb{E}_{\mathbf{p}_{z \oplus i}}[W(y \mid X)]}}\Big)^2 \mathrm{d}\mu$$

$$\leq \frac{1}{2}\int_{\mathcal{Y}} \frac{(\mathbb{E}_{\mathbf{p}_z}[W(y \mid X)] - \mathbb{E}_{\mathbf{p}_{z \oplus i}}[W(y \mid X)])^2}{\mathbb{E}_{\mathbf{p}_z}[W(y \mid X)]} \mathrm{d}\mu. \tag{33}$$

Moreover, under Assumption 1; for any $W \in \mathcal{W}_t$ and $y \in \mathcal{Y}$,

$$\mathbb{E}_{\mathbf{p}_{z \oplus i}}[W(y \mid X)] = \mathbb{E}_{\mathbf{p}_z}\Big[\frac{\mathrm{d}\mathbf{p}_{z \oplus i}}{\mathrm{d}\mathbf{p}_z} \cdot W(y \mid X)\Big] = \mathbb{E}_{\mathbf{p}_z}[(1 + \phi_{z,i}(X)) \cdot W(y \mid X)].$$

Plugging this back into (33), we get

$$\mathrm{d}_{\mathrm{H}}\big(\mathbf{p}_z^W, \mathbf{p}_{z \oplus i}^W\big)^2 \leq \frac{1}{2}\int_{\mathcal{Y}} \frac{\mathbb{E}_{\mathbf{p}_z}[\phi_{z,i}(X)W(y \mid X)]^2}{\mathbb{E}_{\mathbf{p}_z}[W(y \mid X)]} \mathrm{d}\mu.$$

Combining this with Lemma 7 concludes the proof of Theorem 1.

**Proof of Lemma 7.** Our first step is to use the Cauchy–Schwarz inequality, followed by an inequality relating total variation and Hellinger distances:

$$
\frac{1}{k}\left(\sum_{i=1}^{k}\mathrm{d}_{\mathrm{TV}}\left(\mathbf{p}_{+i}^{Y^n},\mathbf{p}_{-i}^{Y^n}\right)\right)^2 \leq \sum_{i=1}^{k}\mathrm{d}_{\mathrm{TV}}\left(\mathbf{p}_{+i}^{Y^n},\mathbf{p}_{-i}^{Y^n}\right)^2
$$

$$
\leq 2\sum_{i=1}^{k}\mathrm{d}_{\mathrm{H}}\left(\mathbf{p}_{+i}^{Y^n},\mathbf{p}_{-i}^{Y^n}\right)^2
$$

$$
\leq 2\sum_{i=1}^{k}\mathbb{E}_Z\left[\mathrm{d}_{\mathrm{H}}\left(\mathbf{p}_Z^{Y^n},\mathbf{p}_{Z\oplus i}^{Y^n}\right)^2 \mid Z_i = +1\right]
$$

$$
= 2\sum_{i=1}^{k}\mathbb{E}_Z\left[\mathrm{d}_{\mathrm{H}}\left(\mathbf{p}_Z^{Y^n},\mathbf{p}_{Z\oplus i}^{Y^n}\right)^2\right], \tag{34}
$$

where the last inequality uses joint convexity of squared Hellinger distance, and the final identity is due to independence of each coordinate of $Z$ and symmetry of Hellinger whereby $\mathbb{E}_Z\left[\mathrm{d}_{\mathrm{H}}\left(\mathbf{p}_Z^{Y^n},\mathbf{p}_{Z\oplus i}^{Y^n}\right)^2 \mid Z_i = +1\right] = \mathbb{E}_Z\left[\mathrm{d}_{\mathrm{H}}\left(\mathbf{p}_Z^{Y^n},\mathbf{p}_{Z\oplus i}^{Y^n}\right)^2 \mid Z_i = -1\right]$.

In order to bound the resulting terms of the sum, we will rely on the so-called *cut-paste* property of Hellinger distance [6]. Before doing so, we will require an additional piece of notation: for fixed $z \in \mathcal{Z}, i \in [k], t \in [n]$, let $\mathbf{p}_{t\leftarrow z\oplus i}^{Y^n}$ denote the message distribution where player $t$ gets a sample from $\mathbf{p}_{z\oplus i}$ and all other players get samples from $\mathbf{p}_z$. That is, for all $y^n \in \mathcal{Y}^n$, the density of $\mathbf{p}_{t\leftarrow z\oplus i}^{Y^n}$ with respect to the underlying product measure $\mu^{\otimes n}$ is given by

$$
\frac{\mathrm{d}\mathbf{p}_{t\leftarrow z\oplus i}^{Y^n}}{\mathrm{d}\mu^{\otimes n}}(y^n) = \mathbb{E}_{X_t\sim\mathbf{p}_{z\oplus i}}\left[W^{y^{t-1}}(y_t \mid X_t)\right] \cdot \prod_{j\neq t}\mathbb{E}_{X_j\sim\mathbf{p}_z}\left[W^{y^{j-1}}(y_j \mid X_j)\right]. \tag{35}
$$

The following lemma, due to [22], allows us to relate $\mathrm{d}_{\mathrm{H}}\left(\mathbf{p}_z^{Y^n},\mathbf{p}_{z\oplus i}^{Y^n}\right)$, the distance between message distributions when all players get observations from $\mathbf{p}_z$, or all from $\mathbf{p}_{z\oplus i}$, to the distances $\mathrm{d}_{\mathrm{H}}\left(\mathbf{p}_z^{Y^n},\mathbf{p}_{t\leftarrow z\oplus i}^{Y^n}\right)$ where only *one* of the $n$ players gets a sample from $\mathbf{p}_{z\oplus i}$.

**Lemma 8** ([22, Theorem 7]). *There exists $c_{\mathrm{H}} > 0$ such that for all $z \in \mathcal{Z}$ and $i \in [k]$,*

$$
\mathrm{d}_{\mathrm{H}}\left(\mathbf{p}_z^{Y^n},\mathbf{p}_{z\oplus i}^{Y^n}\right)^2 \leq c_{\mathrm{H}}\sum_{t=1}^{n}\mathrm{d}_{\mathrm{H}}\left(\mathbf{p}_z^{Y^n},\mathbf{p}_{t\leftarrow z\oplus i}^{Y^n}\right)^2.
$$

*Moreover, one can take $c_{\mathrm{H}} = 2\prod_{t=1}^{\infty}\frac{1}{1-2^{-t}} < 7$.*

Combining Eq. (34) and Lemma 8, we get

$$
\frac{1}{k}\left(\sum_{i=1}^{k}\mathrm{d}_{\mathrm{TV}}\left(\mathbf{p}_{+i}^{Y^n},\mathbf{p}_{-i}^{Y^n}\right)\right)^2 \leq 14\sum_{i=1}^{k}\sum_{t=1}^{n}\mathbb{E}_Z\left[\mathrm{d}_{\mathrm{H}}\left(\mathbf{p}_Z^{Y^n},\mathbf{p}_{t\leftarrow Z\oplus i}^{Y^n}\right)^2\right]
$$

$$
= 14\sum_{t=1}^{n}\mathbb{E}_Z\left[\sum_{i=1}^{k}\mathrm{d}_{\mathrm{H}}\left(\mathbf{p}_Z^{Y^n},\mathbf{p}_{t\leftarrow Z\oplus i}^{Y^n}\right)^2\right]. \tag{36}
$$

In view of bounding the RHS of (36) term by term, fix $j \in [n]$ and $z \in \mathcal{Z}$. Recalling the expression of $\mathbf{p}_{t \leftarrow z \oplus i}^{Y^n}$ from (35), unrolling the definition of Hellinger distance, and recalling (35), we have

$$2 \sum_{i=1}^{k} d_{\mathrm{H}}\left(\mathbf{p}_z^{Y^n}, \mathbf{p}_{t \leftarrow z \oplus i}^{Y^n}\right)^2$$

$$= \sum_{i=1}^{k} \int_{\mathcal{Y}^n} \left(\sqrt{\frac{d\mathbf{p}_z^{Y^n}}{d\mu^{\otimes n}}} - \sqrt{\frac{d\mathbf{p}_{t \leftarrow z \oplus i}^{Y^n}}{d\mu^{\otimes n}}}\right)^2 d\mu^{\otimes n}$$

$$= \sum_{i=1}^{k} \int_{\mathcal{Y}^n} \prod_{j \neq t} \mathbb{E}_{\mathbf{p}_z}\left[W^{y^{j-1}}(y_j \mid X)\right] \underbrace{\left(\sqrt{\mathbb{E}_{\mathbf{p}_z}\left[W^{y^{t-1}}(y_t \mid X)\right]} - \sqrt{\mathbb{E}_{\mathbf{p}_{z \oplus i}}\left[W^{y^{t-1}}(y_t \mid X)\right]}\right)^2}_{:=f_{i,t}(y^{t-1}, y_t)} d\mu^{\otimes n}$$

$$= \sum_{i=1}^{k} \int_{\mathcal{Y}^{t-1}} \prod_{j<t} \mathbb{E}_{\mathbf{p}_z}\left[W^{y^{j-1}}(y_j \mid X)\right] \int_{\mathcal{Y}} f_{i,t}(y^{t-1}, y_t) \int_{\mathcal{Y}^{n-t}} \prod_{j>t} \mathbb{E}_{\mathbf{p}_z}\left[W^{y^{j-1}}(y_j \mid X)\right] d\mu^{\otimes(t-1)} \, d\mu \, d\mu^{\otimes(n-t)}$$

$$= \sum_{i=1}^{k} \int_{\mathcal{Y}^{t-1}} \prod_{j<t} \mathbb{E}_{\mathbf{p}_z}\left[W^{y^{j-1}}(y_j \mid X)\right] \int_{\mathcal{Y}} f_{i,t}(y^{t-1}, y_t) \left(\int_{\mathcal{Y}^{n-t}} \prod_{j>t} \mathbb{E}_{\mathbf{p}_z}\left[W^{y^{j-1}}(y_j \mid X)\right] d\mu^{\otimes(n-t)}\right) d\mu^{\otimes(t-1)} \, d\mu$$

$$= \sum_{i=1}^{k} \int_{\mathcal{Y}^{t-1}} \prod_{j<t} \mathbb{E}_{\mathbf{p}_z}\left[W^{y^{j-1}}(y_j \mid X)\right] \int_{\mathcal{Y}} f_{i,t}(y^{t-1}, y_t) \, d\mu \, d\mu^{\otimes(t-1)}$$

$$= \int_{\mathcal{Y}^{t-1}} \prod_{j<t} \mathbb{E}_{\mathbf{p}_z}\left[W^{y^{j-1}}(y_j \mid X)\right] \sum_{i=1}^{k} \int_{\mathcal{Y}} f_{i,t}(y^{t-1}, y_t) \, d\mu \, d\mu^{\otimes(t-1)} \ ,$$

where the second-to-last identity uses the observation that, for any fixed $y^t \in \mathcal{Y}^t$,

$$\int_{\mathcal{Y}^{n-t}} \prod_{j>t} \mathbb{E}_{\mathbf{p}_z}\left[W^{y^{j-1}}(y_j \mid X)\right] d\mu^{\otimes(n-t)} = 1,$$

which in turn follows upon taking marginal integrals for each coordinate. We then get from the pointwise inequality $\sum_{i=1}^{k} \int_{\mathcal{Y}^{t-1}} f_{i,t}(y^{t-1}, y_t) \, d\mu \leq \sup_{y' \in \mathcal{Y}^{t-1}} \sum_{i=1}^{k} \int_{\mathcal{Y}} f_{i,t}(y', y_t) \, d\mu$ that

$$2 \sum_{i=1}^{k} d_{\mathrm{H}}\left(\mathbf{p}_z^{Y^n}, \mathbf{p}_{t \leftarrow z \oplus i}^{Y^n}\right)^2 \leq \int_{\mathcal{Y}^{t-1}} \prod_{j<t} \mathbb{E}_{\mathbf{p}_z}\left[W^{y^{j-1}}(y_j \mid X)\right] \sup_{y' \in \mathcal{Y}^{t-1}} \sum_{i=1}^{k} \left(\int_{\mathcal{Y}} f_{i,t}(y', y_t) \, d\mu\right) d\mu^{\otimes(t-1)}$$

$$= \left(\sup_{y' \in \mathcal{Y}^{t-1}} \sum_{i=1}^{k} \int_{\mathcal{Y}} f_{i,t}(y', y_t) \, d\mu\right) \int_{\mathcal{Y}^{t-1}} \prod_{j<t} \mathbb{E}_{\mathbf{p}_z}\left[W^{y^{j-1}}(y_j \mid X)\right] d\mu^{\otimes(t-1)}$$

$$= \sup_{y' \in \mathcal{Y}^{t-1}} \sum_{i=1}^{k} \int_{\mathcal{Y}} \left(\sqrt{\mathbb{E}_{\mathbf{p}_z}[W^{y'}(y \mid X)]} - \sqrt{\mathbb{E}_{\mathbf{p}_{z \oplus i}}[W^{y'}(y \mid X)]}\right)^2 d\mu$$

$$\leq \sup_{W \in \mathcal{W}_t} \sum_{i=1}^{k} \int_{\mathcal{Y}} \left(\sqrt{\mathbb{E}_{\mathbf{p}_z}[W(y \mid X)]} - \sqrt{\mathbb{E}_{\mathbf{p}_{z \oplus i}}[W(y \mid X)]}\right)^2 d\mu$$

$$= 2 \cdot \sup_{W \in \mathcal{W}_t} \sum_{i=1}^{k} d_{\mathrm{H}}\left(\mathbf{p}_z^W, \mathbf{p}_{z \oplus i}^W\right)^2. \tag{37}$$

the second identity follows upon taking marginal integrals, and by replacing $f_{i,t}$ by its definition; and the second inequality using that $\left\{ W^{y'} : y' \in \mathcal{Y}^{t-1} \right\} \subseteq \mathcal{W}_t$, so that we are taking a supremum over a larger set.

Plugging this back into (36) and upper bounding the inner expectation by a maximum concludes the proof of the lemma. $\qquad\square$

## E.2 Proof of Theorem 2

Our starting point is Eq. (3) which holds under Assumption 1. We will bound the right-hand-side of Eq. (3) under assumptions of orthogonality and subgaussianity to prove the two bounds in Theorem 2.

First, under orthogonality (Assumption 2), we apply Bessel's inequality to Eq. (3). For a fixed $z \in \mathcal{Z}$, write $\psi_{z,i} = \frac{\phi_{z,i}}{\sqrt{\mathbb{E}_{\mathbf{p}_z}[\phi_{z,i}^2]}}$, and complete $(1, \psi_{z,1}, \ldots, \psi_{z,k})$ to get an orthonormal basis $\mathcal{B}$ for $L^2(\mathcal{X}, \mathbf{p}_z)$. Fix any $W \in \mathcal{W}$ and $y \in \mathcal{Y}$, and, for brevity, define $a \colon \mathcal{X} \to \mathbb{R}$ as $a(x) = W(y \mid x)$. Then, we have

$$\sum_{i=1}^{k} \mathbb{E}[\phi_{z,i}(X)a(X)]^2 \leq \alpha^2 \sum_{i=1}^{k} \mathbb{E}[\psi_{z,i}(X)a(X)]^2 = \alpha^2 \sum_{i=1}^{k} \langle a, \psi_{z,i} \rangle^2 = \alpha^2 \sum_{i=1}^{k} \langle a - \mathbb{E}[a], \psi_{z,i} \rangle^2$$
$$\leq \alpha^2 \sum_{\psi \in \mathcal{B}} \langle a - \mathbb{E}[a], \psi \rangle^2 = \alpha^2 \operatorname{Var}[a(X)],$$

where for the second identity we used the assumption that $\langle \mathbb{E}[a], \psi_{z,i} \rangle = 0$ for all $i \in [k]$ (since 1 and $\psi_{z,i}$ are orthogonal). This establishes Eq. (4).

Turning to Eq. (5), suppose that Assumption 3 holds. Fix $z \in \mathcal{Z}$, and consider any $W \in \mathcal{W}$ and $y \in \mathcal{Y}$. Upon applying Lemma 4 of the Supplement (See Supplement (Appendix B) for the precise statement and proof) to the $\sigma^2$-subgaussian random vector $\phi_z(X)$ and with $a(x)$ set to $W(y \mid x) \in [0, 1]$, we get that

$$\sum_{i=1}^{k} \mathbb{E}_{\mathbf{p}_z}[\phi_{z,i}(X)W(y \mid X)]^2 = \|\mathbb{E}_{\mathbf{p}_z}[\phi_z(X)W(y \mid X)]\|_2^2$$
$$\leq 2\sigma^2 \mathbb{E}_{\mathbf{p}_z}[W(y \mid X)] \cdot \mathbb{E}_{\mathbf{p}_z}\left[W(y \mid X) \log \frac{W(y \mid X)}{\mathbb{E}_{\mathbf{p}_z}[W(y \mid X)]}\right]$$

Integrating over $y \in \mathcal{Y}$, this gives

$$\int_{\mathcal{Y}} \frac{\sum_{i=1}^{k} \mathbb{E}_{\mathbf{p}_z}[\phi_{z,i}(X)W(y \mid X)]^2}{\mathbb{E}_{\mathbf{p}_z}[W(y \mid X)]} \, \mathrm{d}\mu \leq 2\sigma^2 \cdot \int_{\mathcal{Y}} \mathbb{E}_{\mathbf{p}_z}\left[W(y \mid X) \log \frac{W(y \mid X)}{\mathbb{E}_{\mathbf{p}_z}[W(y \mid X)]}\right] \mathrm{d}\mu$$
$$= 2\sigma^2 I(\mathbf{p}_z; W),$$

which yields the claimed bound.

## E.3 Proof of Corollary 1

For any $W \in \mathcal{W}^{\mathrm{priv}, \varepsilon}$, the $\varepsilon$-LDP condition from Eq. (2) can be seen to imply that, for every $y \in \mathcal{Y}$,

$$W(y \mid x_1) - W(y \mid x_2) \leq (e^\varepsilon - 1)W(y \mid x_3), \qquad \forall x_1, x_2, x_3 \in \mathcal{X}.$$

By taking expectation over $x_3$ then again either over $x_1$ or $x_2$ (all distributed according to $\mathbf{p}_z$), this yields

$$|W(y \mid x) - \mathbb{E}_{\mathbf{p}_z}[W(y \mid X)]| \leq (e^\varepsilon - 1)\mathbb{E}_{\mathbf{p}_z}[W(y \mid X)], \qquad \forall x \in \mathcal{X}.$$

Squaring and taking the expectation on both sides, we obtain

$$\operatorname{Var}_{\mathbf{p}_z}[W(y \mid X)] \leq (e^\varepsilon - 1)^2 \, \mathbb{E}_{\mathbf{p}_z}[W(y \mid X)]^2.$$

Dividing by $\mathbb{E}_{\mathbf{p}_z}[W(y \mid X)]$, summing over $y \in \mathcal{Y}$, and using $\int_{\mathcal{Y}} \mathbb{E}_{\mathbf{p}_z}[W(y \mid X)] \, \mathrm{d}\mu = 1$ gives

$$\int_{\mathcal{Y}} \frac{\operatorname{Var}_{\mathbf{p}_z}[W(y \mid X)]}{\mathbb{E}_{\mathbf{p}_z}[W(y \mid X)]} \, \mathrm{d}\mu \leq (e^\varepsilon - 1)^2 \int_{\mathcal{Y}} \mathbb{E}_{\mathbf{p}_z}[W(y \mid X)] \, \mathrm{d}\mu = (e^\varepsilon - 1)^2,$$

thus establishing (6). For the bound of $e^\varepsilon$, observe that, for all $y \in \mathcal{Y}$,

$$\operatorname{Var}_{\mathbf{p}_z}[W(y \mid X)] \leq \mathbb{E}_{\mathbf{p}_z}[W(y \mid X)^2] \leq e^\varepsilon \min_{x \in \mathcal{X}} W(y \mid x) \mathbb{E}_{\mathbf{p}_z}[W(y \mid X)].$$

Hence

$$\int_{\mathcal{Y}} \frac{\mathrm{Var}_{\mathbf{p}_z}[W(y \mid X)]}{\mathbb{E}_{\mathbf{p}_z}[W(y \mid X)]} \, \mathrm{d}\mu \le e^{\varepsilon} \int_{\mathcal{Y}} \min_{x \in \mathcal{X}} W(y \mid x) \, \mathrm{d}\mu \le e^{\varepsilon} \cdot \min_{x \in \mathcal{X}} \int_{\mathcal{Y}} W(y \mid x) \, \mathrm{d}\mu = e^{\varepsilon}.$$

The bound (7) (under Assumption 3) will follow from (5), and the relation between differential privacy and KL divergence. Indeed, the mutual information $I(\mathbf{p}_z; W)$ can be rewritten as the expected (over $X \sim \mathbf{p}_Z$) KL divergence between the distribution $\mathbf{p}^{W,X} := W(\cdot \mid X)$ over $\mathcal{Y}$ induced by the channel $W$ on input $X$, and the distribution $\mathbf{p}_Z^W := \mathbb{E}_{X' \sim \mathbf{p}_z}[W(\cdot \mid X')]$ over $\mathcal{Y}$ induced by the input distribution $\mathbf{p}_z$ and the channel $W$:

$$I(\mathbf{p}_z; W) = \mathbb{E}_{X \sim \mathbf{p}_z}\left[\mathrm{D}\big(\mathbf{p}^{W,X} \| \mathbf{p}_z^W\big)\right] = \mathbb{E}_{X \sim \mathbf{p}_z}\left[\mathbb{E}_{Y \sim \mathbf{p}^{W,X}}\left[\ln \frac{W(Y \mid X)}{\mathbb{E}_{X' \sim \mathbf{p}_z}[W(Y \mid X')]}\right]\right];$$

but the $\varepsilon$-LDP condition from Eq. (2) guarantees that the log-likelihood ratio in the inner expectation is (almost surely) at most $\varepsilon$, so that $I(\mathbf{p}_z; W) \le \varepsilon$ for every $z$ and $W \in \mathcal{W}^{\mathrm{priv},\varepsilon}$. This yields (7).

### E.4 Proof of Corollary 2

In view of (4), to establish (8), it suffices to show that $\frac{\mathrm{Var}_{\mathbf{p}_z}[W(y|X)]}{\mathbb{E}_{\mathbf{p}_z}[W(y|X)]} \le 1$ for every $y \in \mathcal{Y}$. Since $W(y \mid x) \in (0,1]$ for all $x \in \mathcal{X}$ and $y \in \mathcal{Y}$, so that

$$\mathrm{Var}_{\mathbf{p}_z}[W(y \mid X)] \le \mathbb{E}_{\mathbf{p}_z}\big[W(y \mid X)^2\big] \le \mathbb{E}_{\mathbf{p}_z}[W(y \mid X)].$$

The second bound (under Assumption 3) will follow from (5). Indeed, recalling that the entropy of the output of a channel is bounded below by the mutual information between input and the output, we have $I(\mathbf{p}_z; W) \le H(\mathbf{p}_z^W)$, where $\mathbf{p}_z^W := \mathbb{E}_{\mathbf{p}_z}[W(\cdot \mid X)]$ is the distribution over $\mathcal{Y}$ induced by the input distribution $\mathbf{p}_z$ and the channel $W$. Using the fact that the entropy of a distribution over $\mathcal{Y}$ is at most $\log |\mathcal{Y}|$ in (5) gives (9).

## F   Missing proofs in Section 4

### F.1   Proof of Lemma 1

Given an $(n, \gamma)$-estimator $(\Pi, \hat{\theta})$, define an estimate $\hat{Z}$ for $Z$ as

$$\hat{Z} := \operatorname*{argmin}_{z \in \mathcal{Z}} \left\| \theta_z - \hat{\theta}(Y^n, U) \right\|_p.$$

By the triangle inequality,

$$\left\| \theta_Z - \theta_{\hat{Z}} \right\|_p \le \left\| \theta_Z - \hat{\theta}(Y^n, U) \right\|_p + \left\| \theta_{\hat{Z}} - \hat{\theta}(Y^n, U) \right\|_p \le 2 \left\| \hat{\theta}(Y^n, U) - \theta_Z \right\|_p.$$

Since $(\Pi, \hat{\theta})$ is an $(n, \gamma)$-estimator under $\ell_p$ loss for $\mathcal{P}_\Theta$,

$$\mathbb{E}_Z\left[\mathbb{E}_{\mathbf{p}_z}\left[\left\| \theta_Z - \theta_{\hat{Z}} \right\|_p^p\right]\right] \le 2^p \gamma^p \Pr[\mathbf{p}_Z \in \mathcal{P}_\Theta] + \max_{z \ne z'} \|\theta_z - \theta_{z'}\|_p^p \Pr[\mathbf{p}_Z \notin \mathcal{P}_\Theta]$$

$$\le 2^p \gamma^p + 4^p \gamma^p \frac{1}{\tau} \cdot \frac{\tau}{4} \tag{38}$$

$$\le \frac{3}{4} 4^p \gamma^p, \tag{39}$$

where Eq. (38) follows from Assumption 4 and $\Pr[\mathbf{p}_Z \in \mathcal{P}_\Theta] \ge 1 - \tau/4$. Next, for $p \in [1, \infty)$, by Assumption 4, $\|\theta_Z - \theta_{\hat{Z}}\|_p^p \ge \frac{4^p \gamma^p}{\tau k} \sum_{i=1}^k \mathbb{1}\big\{Z_i \ne \hat{Z}_i\big\}$. Combining with Eq. (39) this shows that $\frac{1}{\tau k} \sum_{i=1}^k \Pr\big[Z_i \ne \hat{Z}_i\big] \le \frac{3}{4}$.

Furthermore, since the Markov relation $Z_i - (Y^n, U) - \hat{Z}_i$ holds for all $i$, we can lower bound $\Pr\big[Z_i \ne \hat{Z}_i\big]$ using the standard relation between total variation distance and hypothesis testing as

follows, using that $\tau \leq 1/2$ in the second inequality:

$$\Pr\Big[ Z_i \neq \hat{Z}_i \Big] \geq \tau \Pr\Big[ \hat{Z}_i = -1 \;\Big|\; Z_i = 1 \Big] + (1-\tau)\Pr\Big[ \hat{Z}_i = 1 \;\Big|\; Z_i = -1 \Big]$$

$$\geq \tau\Big( \Pr\Big[ \hat{Z}_i = -1 \;\Big|\; Z_i = 1 \Big] + \Pr\Big[ \hat{Z}_i = 1 \;\Big|\; Z_i = -1 \Big] \Big)$$

$$\geq \tau\Big( 1 - \mathrm{d}_{\mathrm{TV}}\Big( \mathbf{p}_{+i}^{Y^n}, \mathbf{p}_{-i}^{Y^n} \Big) \Big).$$

Summing over $1 \leq i \leq k$ and combining it with the previous bound, we obtain

$$\frac{3}{4} \geq \frac{1}{\tau k}\sum_{i=1}^{k}\Pr\Big[ Z_i \neq \hat{Z}_i \Big] \geq 1 - \frac{1}{k}\sum_{i=1}^{k}\mathrm{d}_{\mathrm{TV}}\Big( \mathbf{p}_{+i}^{Y^n}, \mathbf{p}_{-i}^{Y^n} \Big)$$

and reorganizing proves the result.

## G  Missing statements and proofs in Section 5

### G.1  Proof of Theorem 3

Fix $p \in [1, \infty)$. Let $k = d$, $\mathcal{Z} = \{-1, +1\}^d$, and $\tau = \frac{s}{2d}$; and suppose that, for some $\gamma \in (0, 1/8]$, there exists an $(n, \gamma)$-estimator for $\mathcal{B}_{d,s}$ under $\ell_p$ loss. We fix a parameter $\gamma \in (0, 1/2]$, which will be chosen as a function of $\gamma, d, p$ later. Consider the set of $2^d$ product Bernoulli distributions $\{\mathbf{p}_z\}_{z \in \mathcal{Z}}$, where $\mu(\mathbf{p}_z) = \mu_z := \frac{1}{2}\gamma(z + \mathbf{1}_d)$ (so the sparsity of the mean vector is equal to the number of positive coordinates of $z$). We have, for $z \in \mathcal{Z}$,

$$\mathbf{p}_z(x) = \frac{1}{2^d}\prod_{i=1}^{d}\Big( 1 + \frac{1}{2}\gamma(z_i + 1)x_i \Big), \qquad x \in \mathcal{X}.$$

It follows for $z \in \mathcal{Z}$ and $i \in [d]$ that

$$\mathbf{p}_{z \oplus i}(x) = \frac{1 + \frac{1}{2}\gamma(1 - z_i)x_i}{1 + \frac{1}{2}\gamma(1 + z_i)x_i}\mathbf{p}_z(x) = \Big( 1 - \gamma\frac{z_i x_i}{1 + \frac{1}{2}\gamma(1 + z_i)x_i} \Big)\mathbf{p}_z(x) = (1 + \phi_{z,i}(x))\mathbf{p}_z(x) \tag{40}$$

where $\phi_{z,i}(x) := -\frac{\gamma z_i x_i}{1 + \frac{1}{2}\gamma(1 + z_i)x_i}$. We can verify that, for $i \neq j$,

$$\mathbb{E}_{\mathbf{p}_z}[\phi_{z,i}(X)] = 0, \quad \mathbb{E}_{\mathbf{p}_z}\big[\phi_{z,i}(X)^2\big] = \frac{\gamma^2}{1 - \frac{1}{2}\gamma^2(1 + z_i)}, \text{ and } \mathbb{E}_{\mathbf{p}_z}[\phi_{z,i}(X)\phi_{z,j}(X)] = 0,$$

so that Assumptions 1 and 2 are satisfied for $\alpha^2 := 2\gamma^2$. Moreover, using, *e.g.*, Hoeffding's lemma (*cf.* [9]), for $\gamma < 1$, the random vector $\phi_z(X) = (\phi_{z,i}(X))_{i \in [d]}$ is $\frac{\gamma^2}{(1-\gamma^2)^2}$-subgaussian. Thus, Assumption 3 holds as well, and we can invoke both parts of Theorem 2.

Let $\|z\|_+ := |\{i \in [d] \mid z_i = 1\}|$, so that $\|\mu_z\|_0 = \sum_{i=1}^{d}\frac{1}{2}(1 + z_i) = \|z\|_+$. The next claim, which follows from standard bounds for binomial random variables, states that when $Z \sim \mathrm{Rad}(\tau)^{\otimes d}$, $\mu_Z$ is $s$-sparse with high probability.

**Fact 2.** *Let $Z \sim \mathrm{Rad}(\tau)^{\otimes d}$, where $\tau d \geq 4\log d$. Then $\Pr\big[ \|Z\|_+ \leq 2\tau d \big] \geq 1 - \tau/4$.*

Hence the construction satisfies $\Pr_Z[\mathbf{p}_Z \in \mathcal{B}_{d,s}] \leq 1 - \tau/4$, as required in Lemma 1.

We now choose $\gamma = \gamma(p) := \frac{4\gamma}{(s/2)^{1/p}} \in (0, 1/2]$, which implies that Assumption 4 holds since

$$\ell_p(\mu(\mathbf{p}_z), \mu(\mathbf{p}_{z'})) = \gamma\, \mathrm{d}_{\mathrm{Ham}}(z, z')^{1/p} = 4\gamma\Big( \frac{\mathrm{d}_{\mathrm{Ham}}(z, z')}{\tau d} \Big)^{1/p}.$$

Therefore, we can apply Lemma 1 as well. For $\mathcal{W}^{\mathrm{priv},\varepsilon}$, we prove the two parts of the lower bound separately, depending on whether $\varepsilon \leq 1$. First, upon combining the bounds obtained by Corollary 1 and Lemma 1 (specifically, for the former, (6)), we get

$$d \leq 112 n \alpha^2 (e^\varepsilon - 1)^2,$$

whereby, upon recalling that $\alpha^2 = 2\gamma^2$, and using the value of $\gamma = \gamma(p)$ above, it follows that

$$\frac{1}{3584} \cdot \frac{d(s/2)^{\frac{2}{p}}}{n(e^\varepsilon - 1)^2} \leq \gamma^2.$$

Thus, $\mathcal{E}_p(\mathcal{B}_{d,s}, \mathcal{W}^{\mathrm{priv},\varepsilon}, n) = \Omega\left(\sqrt{\frac{ds^{2/p}}{n\varepsilon^2}}\right)$ for $\varepsilon \in (0, 1]$. For the second part of the bound, which dominates for $\varepsilon > 1$, observe that Assumption 3 holds with $\sigma^2 := \frac{\gamma^2}{(1-\gamma^2)^2} \leq 2\gamma^2$; allowing us to apply the second part of Corollary 1, (7), which as before combined with Lemma 1 yields

$$d \leq 224n\sigma^2\varepsilon \leq 448n\gamma^2\varepsilon,$$

and again from the setting of $\gamma$ we get $\mathcal{E}_p(\mathcal{B}_{d,s}, \mathcal{W}^{\mathrm{priv},\varepsilon}, n) = \Omega\left(\sqrt{\frac{ds^{2/p}}{n\varepsilon}}\right)$.

Similarly, for $\mathcal{W}^{\mathrm{comm},\ell}$, again since Assumption 3 holds with $\sigma^2 \leq 2\gamma^2$, upon combining the bounds obtained by Corollary 2 and Lemma 1, we get

$$\frac{ds^{\frac{2}{p}}}{28672n\ell} \leq \gamma^2,$$

which gives $\mathcal{E}_p(\mathcal{B}_{d,s}, \mathcal{W}^{\mathrm{comm},\ell}, n) = \Omega\left(\sqrt{\frac{ds^{2/p}}{n\ell}} \wedge 1\right)$. Finally, note that for $\ell \geq d$, the lower bound follows from the minimax rate in the unconstrained setting, which can be seen to be $\Omega\left(\sqrt{s^{2/p}\log(2d/s)/n}\right)$ [28, 30]. This completes the proof.

This handles the case $p \in [1, \infty)$. For $p = \infty$, the lower bounds immediately follow from plugging $p = \log s$ in the previous expressions, as discussed in Footnote 3.

### G.2 Proof of Theorem 4

We denote the mean by $\mu$ instead of $\theta$, denote the estimator by $\hat{\mu}$, and consider the minimax error rate $\mathcal{E}_p(\mathcal{G}_{d,s}, \mathcal{W}, n)$ of mean estimation for $\mathcal{P}_\Theta = \mathcal{G}_{d,s}$ using $\mathcal{W}$ under $\ell_p$ loss.

*Proof of Theorem 4.* Let $\varphi$ denote the probability density function of the standard Gaussian distribution $\mathcal{G}(\mathbf{0}, \mathbb{I})$. Fix $p \in [1, \infty)$. Let $k = d$, $\mathcal{Z} = \{-1, +1\}^d$, and $\tau = \frac{s}{2d}$; and suppose that, for some $\gamma \in (0, 1/8]$, there exists an $(n, \gamma)$-estimator for $\mathcal{G}_{d,s}$ under $\ell_p$ loss. We fix a parameter $\gamma := \gamma(p) := \frac{4\gamma}{(s/2)^{1/p}} \in (0, 1/2]$, and consider the set of distributions $\{\mathbf{p}_z\}_{z \in \mathcal{Z}}$ of all $2^d$ spherical Gaussian distributions with mean $\mu_z := \gamma(z + \mathbf{1}_d)$, where $z \in \mathcal{Z}$. Again, note that $\|\mu_z\|_0 = \sum_{i=1}^d \mathbb{1}\{z_i = 1\} = \|z\|_+$, and Fact 2 applies here too. Then by the definition of Gaussian density, for $z \in \mathcal{Z}$,

$$\mathbf{p}_z(x) = e^{-\gamma^2\|\mu_z\|_2^2/2} \cdot e^{\gamma\langle x, z + \mathbf{1}_d\rangle} \cdot \varphi(x). \tag{41}$$

Therefore, for $z \in \mathcal{Z}$ and $i \in [d]$, we have

$$\mathbf{p}_{z \oplus i}(x) = e^{-2\gamma x_i z_i}e^{2\gamma^2 z_i} \cdot \mathbf{p}_z(x) = (1 + \phi_{z,i}(x)) \cdot \mathbf{p}_z(x), \tag{42}$$

where $\phi_{z,i}(x) := 1 - e^{-2\gamma x_i z_i}e^{2\gamma^2 z_i}$. By using the Gaussian moment-generating function, for $i \neq j$,

$$\mathbb{E}_{\mathbf{p}_z}[\phi_{z,i}(X)] = 0, \quad \mathbb{E}_{\mathbf{p}_z}\left[\phi_{z,i}(X)^2\right] = e^{4\gamma^2} - 1, \text{ and } \mathbb{E}_{\mathbf{p}_z}[\phi_{z,i}(X)\phi_{z,j}(X)] = 0,$$

so that Assumptions 1 and 2 are satisfied for $\alpha^2 := e^{4\gamma^2} - 1$. By our choice of $\gamma$ and the assumption on $\gamma$, one can check that Assumption 4 holds:

$$\ell_p(\mu(\mathbf{p}_z), \mu(\mathbf{p}_{z'})) = 4\gamma\left(\frac{\mathrm{d}_{\mathrm{Ham}}(z, z')}{\tau d}\right)^{1/p}.$$

Moreover, similar to the product of Bernoulli case, using Fact 2, we can show that $\mathrm{Pr}_Z[\mathbf{p}_Z \in \mathcal{G}_{d,s}] \leq 1 - \tau/4$. This allows us to apply Lemma 1.

### G.2.1 Privacy constraints for $\varepsilon \in (0,1)$

For $\mathcal{W}^{\mathrm{priv},\varepsilon}$, upon combining the bounds obtained by Corollary 1 and Lemma 1, we get

$$d \leq 112 n \alpha^2 (e^\varepsilon - 1)^2,$$

whereby, upon noting that $\alpha^2 = e^{4\gamma^2} - 1 \leq 8\gamma^2$ holds since $\gamma \leq 1/2$, and using the value of $\gamma = \gamma(p)$ above, it follows that

$$\gamma^2 \geq \frac{d(s/2)^{\frac{2}{p}}}{14336 \cdot n(e^\varepsilon - 1)^2}.$$

Thus, $\mathcal{E}_p(\mathcal{G}_{d,s}, \mathcal{W}^{\mathrm{priv},\varepsilon}, n) = \Omega\left(\sqrt{\frac{ds^{2/p}}{n\varepsilon^2}} \wedge 1\right)$. This establishes the lower bounds for $\mathcal{W}^{\mathrm{priv},\varepsilon}$. (Recall that the bound for $p = \infty$ then follows from setting $p = \log d$.)

### G.2.2 Communication constraints, and privacy constraints for $\varepsilon \geq 1$

For these cases, to prove a lower bound with the desired dependence on $\varepsilon$ or $\ell$, we will need to use the tighter bounds in Corollaries 1 and 2 which hold only under Assumption 3. This, however, leads to an issue: the random vector $\phi_z(X) = (\phi_{z,i}(X))_{i \in [d]}$ is not subgaussian, due to the one-sided exponential growth, and therefore Assumption 3 does not hold.

To overcome this and still obtain a linear dependence on $\ell$ (or $\varepsilon$) (instead of the suboptimal $2^\ell$ (or $e^\varepsilon$)), we will consider instead the class of "truncated" Gaussian distributions, whose corresponding $\phi$ functions are subgaussian; and argue that these truncated distributions are close enough to the original Gaussian distributions such a lower bound in the truncated case implies one in the original Gaussian case.

In particular, we consider the following collection of truncated Gaussian distributions. For $z \in \mathcal{Z}$, let $\mathbf{p}_z$ be the density function of a spherical Gaussian distribution with mean $\mu_z$ as defined in Eq. (41). For a truncation bound $B$, let $\mathbf{p}_{z,B}$ be the distribution of $X \sim \mathbf{p}_z$ conditioned on the event that $\|X\|_\infty \leq B$. That is, we have, for $x \in \mathbb{R}^d$,

$$\mathbf{p}_{z,B}(x) = C_z \mathbf{p}_z(x) \mathbb{1}\{\|X\|_\infty \leq B\},$$

where $C_z = 1/\Pr_{X \sim \mathbf{p}_z}[\|X\|_\infty \leq B]$. Then the following bound follows from standard Gaussian concentration bound on each dimension and a union bound over all dimensions.

**Fact 3.** *Setting $B := 4\sqrt{\ln(dn)}$, we have, for every $z \in \mathcal{Z}$, $\mathrm{d}_{\mathrm{TV}}(\mathbf{p}_{z,B}, \mathbf{p}_z) \leq \frac{1}{d^7 n^8}$.*

Let $\mathbf{p}_{z,B}^{Y^n}$ be the distribution of the messages obtained by executing the protocol when each user gets a sample from $\mathbf{p}_{z,B}$ and let the corresponding mixtures be denoted by $\mathbf{p}_{+i,B}^{Y^n}$ and $\mathbf{p}_{-i,B}^{Y^n}$. Then we have

$$
\begin{aligned}
\mathrm{d}_{\mathrm{TV}}\left(\mathbf{p}_{+i}^{Y^n}, \mathbf{p}_{-i}^{Y^n}\right) &\leq \mathrm{d}_{\mathrm{TV}}\left(\mathbf{p}_{+i,B}^{Y^n}, \mathbf{p}_{-i,B}^{Y^n}\right) + \mathrm{d}_{\mathrm{TV}}\left(\mathbf{p}_{+i}^{Y^n}, \mathbf{p}_{+i,B}^{Y^n}\right) + \mathrm{d}_{\mathrm{TV}}\left(\mathbf{p}_{-i,B}^{Y^n}, \mathbf{p}_{-i}^{Y^n}\right) \\
&\leq \mathrm{d}_{\mathrm{TV}}\left(\mathbf{p}_{+i,B}^{Y^n}, \mathbf{p}_{-i,B}^{Y^n}\right) + \max_z \left\{\mathrm{d}_{\mathrm{TV}}\left(\mathbf{p}_z^{Y^n}, \mathbf{p}_{z,B}^{Y^n}\right) + \mathrm{d}_{\mathrm{TV}}\left(\mathbf{p}_{z,B}^{Y^n}, \mathbf{p}_z^{Y^n}\right)\right\} \\
&\leq \mathrm{d}_{\mathrm{TV}}\left(\mathbf{p}_{+i,B}^{Y^n}, \mathbf{p}_{-i,B}^{Y^n}\right) + 2\max_z \mathrm{d}_{\mathrm{TV}}\left(\mathbf{p}_{z,B}^{\otimes n}, \mathbf{p}_z^{\otimes n}\right) \\
&\leq \mathrm{d}_{\mathrm{TV}}\left(\mathbf{p}_{+i,B}^{Y^n}, \mathbf{p}_{-i,B}^{Y^n}\right) + 2n\max_z \mathrm{d}_{\mathrm{TV}}(\mathbf{p}_{z,B}, \mathbf{p}_z) \\
&\leq \mathrm{d}_{\mathrm{TV}}\left(\mathbf{p}_{+i,B}^{Y^n}, \mathbf{p}_{-i,B}^{Y^n}\right) + \frac{2}{d^7 n^7}.
\end{aligned}
$$

The third inequality follows from data processing inequality and the fourth inequality follows from subadditivity of TV distance.

Combining this with Lemma 1, for any protocol that correctly learns the Gaussian family, we must have

$$\frac{1}{d} \sum_{i=1}^d \mathrm{d}_{\mathrm{TV}}\left(\mathbf{p}_{+i,B}^{Y^n}, \mathbf{p}_{-i,B}^{Y^n}\right) \geq \frac{1}{8}. \tag{43}$$

Next we show that the $\phi$ functions corresponding to $\mathbf{p}_{z,B}$'s are subgaussian and establish the corresponding upper bounds on the average information bound above. Note that

$$\phi_{z,i}^B(x) := \frac{\mathbf{p}_{z\oplus i}^B(x)}{\mathbf{p}_z^B(x)} - 1 = \frac{C_{z\oplus i}}{C_z} e^{-2\gamma x_i z_i} e^{2\gamma^2 z_i} \mathbb{1}\{\|x\|_\infty \le B\} - 1 \tag{44}$$

By the inequality $|ab - 1| \le |a| \cdot |b - 1| + |a - 1|$, we have have, for all $z \in \mathcal{Z}$,

$$\left|\frac{C_{z\oplus i}}{C_z} - 1\right| \le \frac{1}{C_z}|C_{z\oplus i} - 1| + \left|\frac{1}{C_z} - 1\right| \le \left|\frac{1}{\Pr_{X\sim\mathbf{p}_{z\oplus i}}[\|X\|_\infty \le B]} - 1\right| + \left|\Pr_{X\sim\mathbf{p}_z}[\|X\|_\infty \le B] - 1\right|$$

$$\le \frac{10}{d^7 n^7}.$$

Moreover, for all $z \in \mathcal{Z}$, for $\gamma \le \frac{1}{3B}$,

$$\left|e^{-2\gamma x_i z_i} e^{2\gamma^2 z_i} \mathbb{1}\{\|x\|_\infty \le B\} - 1\right| \le \left|e^{2\gamma^2 + 2\gamma B} - 1\right| \le \left|e^{3\gamma B} - 1\right| \le 6\gamma B. \tag{45}$$

Hence, applying the inequality $|ab - 1| \le |a| \cdot |b - 1| + |a - 1|$ again on Eq. (44), we have for $\gamma \le \frac{1}{3B}$,

$$|\phi_{z,i}^B(x)| \le 12\gamma B + \frac{10}{d^7 n^7}.$$

Thus, we get that for all $z \in \mathcal{Z}, i \in [d]$, $\phi_{z,i}^B$ is subgaussian with proxy $\sigma_B = 12\gamma B + \frac{10}{d^7 n^7}$.

Under communication constraints, applying Corollary 2, we get

$$\left(\frac{1}{d}\sum_{i=1}^d d_{\mathrm{TV}}\left(\mathbf{p}_{+i,B}^{Y^n}, \mathbf{p}_{-i,B}^{Y^n}\right)\right)^2 \le \frac{14}{d}\sigma_B^2 n\ell.$$

To conclude, we observe that by plugging our setting of $\gamma = \gamma(p)$ in the above inequality, we must have

$$\gamma^2 \ge \frac{d(s/2)^{\frac{2}{p}}}{14336 \cdot n \cdot B^2\ell}$$

in order to satisfy Eq. (43), hence proving the desired lower bound. The lower bound for LDP with $\varepsilon > 1$ follows similarly by applying Corollary 1. $\qquad\square$

### G.3 Detailed results for discrete family

We derive a lower bound for $\mathcal{E}_p(\Delta_d, \mathcal{W}, n)$, the minimax rate for discrete density estimation, under local privacy and communication constraints.

**Theorem 6.** *Fix $p \in [1, \infty)$. For $\varepsilon > 0$, and $\ell \ge 1$, we have*

$$\mathcal{E}_p(\Delta_d, \mathcal{W}^{\mathrm{priv},\varepsilon}, n) \gtrsim \sqrt{\frac{d^{2/p}}{n((e^\varepsilon - 1)^2 \wedge e^\varepsilon)} \wedge \left(\frac{1}{n((e^\varepsilon - 1)^2 \wedge e^\varepsilon)}\right)^{\frac{p-1}{p}} \wedge 1} \tag{46}$$

*and*

$$\mathcal{E}_p(\Delta_d, \mathcal{W}^{\mathrm{comm},\ell}, n) \gtrsim \sqrt{\frac{d^{2/p}}{n2^\ell} \wedge \left(\frac{1}{n2^\ell}\right)^{\frac{p-1}{p}} \wedge 1}. \tag{47}$$

In particular, for $n((e^\varepsilon - 1)^2 \wedge e^\varepsilon) \ge d^2$ and $n(2^\ell \wedge d) \ge d^2$, the first term of the corresponding lower bounds dominates. Before turning to the proof of this theorem, we note that Corollary 3 and Corollary 4 are direct corollaries of the theorem.

We now establish Theorem 6.

*Proof of Theorem 6.* Fix $p \in [1, \infty)$, and suppose that, for some $\gamma \in (0, 1/16]$, there exists an $(n, \gamma)$-estimator for $\Delta_d$ under $\ell_p$ loss. Set

$$D := d \wedge \left\lfloor \left(\frac{1}{16\gamma}\right)^{\frac{p}{p-1}} \right\rfloor$$

and assume, without loss of generality, that $D$ is even. By definition, we then have $\gamma \in (0, 1/(16D^{1-1/p})]$ and $D \leq d$; we can therefore restrict ourselves to the first $D$ elements of the domain, embedding $\Delta_D$ into $\Delta_d$, to prove our lower bound.

Let $k = \frac{D}{2}$, $\mathcal{Z} = \{-1, +1\}^{D/2}$, and $\tau = \frac{1}{2}$; and suppose that, for some $\gamma \in (0, 1/(16D^{1-1/p})]$, there exists an $(n, \gamma)$-estimator for $\Delta_D$ under $\ell_p$ loss. (We will use the fact that $\gamma \leq 1/(16D^{1-1/p})$ for Eq. (49) to be a valid distribution with positive mass, as we will need $|\gamma| \leq \frac{1}{D}$; and to bound $\alpha^2$ later on, as we will require $|\gamma| \leq \frac{1}{2D}$.) Define $\gamma = \gamma(p)$ as

$$\gamma(p) := \frac{4 \cdot 2^{1/p}\gamma}{D^{1/p}}, \tag{48}$$

which implies $\gamma \in [0, 1/(2D)]$. Consider the set of $D$-ary distributions $\mathcal{P}^{\gamma}_{\text{Discrete}} = \{\mathbf{p}_z\}_{z \in \mathcal{Z}}$ defined as follows. For $z \in \mathcal{Z}$, and $x \in \mathcal{X} = [D]$

$$\mathbf{p}_z(x) = \begin{cases} \frac{1}{D} + \gamma z_i, & \text{if } x = 2i, \\ \frac{1}{D} - \gamma z_i, & \text{if } x = 2i - 1. \end{cases} \tag{49}$$

For $z \in \mathcal{Z}$ and $i \in [D/2]$, we have

$$\begin{aligned}
\mathbf{p}_{z \oplus i}(x) &= \left(1 - \frac{2D\gamma z_i}{1 + D\gamma z_i}\mathbb{1}\{x = 2i\} + \frac{2D\gamma z_i}{1 - D\gamma z_i}\mathbb{1}\{x = 2i - 1\}\right)\mathbf{p}_z(x) \\
&= (1 + \phi_{z,i}(x))\mathbf{p}_z(x), \tag{50}
\end{aligned}$$

where

$$\phi_{z,i}(x) := z_i \cdot \frac{2D\gamma}{1 - D^2\gamma^2}((1 + D\gamma z_i)\mathbb{1}\{x = 2i - 1\} - (1 - D\gamma z_i)\mathbb{1}\{x = 2i\}).$$

Once again, we can verify that for $i \neq j$

$$\mathbb{E}_{\mathbf{p}_z}[\phi_{z,i}(X)] = 0, \quad \mathbb{E}_{\mathbf{p}_z}[\phi_{z,i}(X)^2] = \frac{8\gamma^2 D}{1 - \gamma^2 D^2}, \text{ and } \mathbb{E}_{\mathbf{p}_z}[\phi_{z,i}(X)\phi_{z,j}(X)] = 0,$$

so that Assumptions 1 and 2 are satisfied for $\alpha^2 := 16\gamma^2 D$ (using that $D\gamma \leq 1/2$ to simplify the bound).[10] Thus, we can invoke the first part of Theorem 2. Note that Assumption 4 holds, since $\ell_p(\mathbf{p}_z, \mathbf{p}_{z'}) = \gamma \, \mathrm{d}_{\text{Ham}}(z, z')^{1/p} = 4\gamma \left(\frac{\mathrm{d}_{\text{Ham}}(z,z')}{\tau D}\right)^{1/p}$. Therefore, we can apply Lemma 1 as well.

For $\mathcal{W}^{\text{priv},\varepsilon}$, by combining the bounds obtained by Corollary 1 and Lemma 1, we get

$$D \leq 56n\alpha^2\big((e^\varepsilon - 1)^2 \wedge e^\varepsilon\big),$$

whereby, upon recalling the value of $\alpha^2$ and using the setting of $\gamma = \gamma(p)$ from Eq. (48), it follows that

$$\gamma^2 \geq \frac{D^{\frac{2}{p}}}{7168 \cdot 2^{2/p} \cdot n((e^\varepsilon - 1)^2 \wedge e^\varepsilon)} \asymp \frac{d^{2/p} \wedge \gamma^{-2/(p-1)}}{n((e^\varepsilon - 1)^2 \wedge e^\varepsilon)}.$$

Thus we obtain the bound Eq. (46) as claimed.

Similarly, for $\mathcal{W}^{\text{comm},\ell}$, upon combining the bounds obtained by Corollary 2 and Lemma 1 and recalling that $|\mathcal{Y}| = 2^\ell$, we get

$$\gamma^2 \geq \frac{D^{\frac{2}{p}}}{7168 \cdot 2^{2/p} \cdot n2^\ell},$$

which gives $\mathcal{E}_p(\Delta_D, \mathcal{W}^{\text{comm},\ell}, n) = \Omega\left(\sqrt{\frac{d^{2/p}}{n2^\ell} \wedge \left(\frac{1}{n2^\ell}\right)^{\frac{p-1}{p}}}\right),$[11] concluding the proof. $\qquad\square$

---

[10]It is worth noting that Assumption 3 will not hold for any useful choice of the subgaussianity parameter.

[11]Finally, note that we could replace the quantity $2^\ell$ above by $2^\ell \wedge d$, or even $2^\ell \wedge D$, as for $2^\ell \geq D$ there is no additional information any player can send beyond the first $\log_2 D$ bits, which encode their full observation. However, this small improvement would lead to more cumbersome expressions, and not make any difference for the main case of interest, $p = 1$.

