# OpenReview forum: "Unified Lower Bounds for Interactive High-dimensional Estimation under Information Constraints"
_NeurIPS.cc/2023/Conference — NeurIPS 2023 poster_

### Official Review · Reviewer_Zd1o · 2023-07-02

**Soundness:** 3 good
**Presentation:** 3 good
**Contribution:** 3 good
**Rating:** 6
**Confidence:** 4

**Summary:**

This paper discusses parametric estimation under a communication setup. This setup adds variation to classic parametric estimation and focuses on the setup $\theta \to X^n \to Y^n$ where the goal is to estimate $\theta$ given $Y^n$, which is generated in an interactive, sequential manner with $Y_i$ possibly dependent on other signals.

The main result of this paper is Theorem 1, which gives an information contraction bound. With additional assumptions on orthogonality and subgaussianity, the paper's next result is Theorem 2, which gives two corollary information contraction bounds expressed in terms of variance, mean and mutual information terms relating to channels.

The authors then provide several examples of tightness (and near tightness) for these bounds by looking at minimax rates for examples such as product Bernoulli, sparse Gaussian, and discrete distributions under information or communication constraints.



**Strengths:**

**Originality**: The paper builds upon the classic framework of parametric estimation under an interactive framework. While the problem setup itself is not original, this paper contributes new findings to minimax theory for these setups, particularly with communication and privacy constraints.

**Quality**: The paper is nicely written, and the authors explain the setting and results well. No glaring issues were found with regards to technicalities.

**Clarity**: Overall, the paper is cleanly written, bar some comments stated in below sections.

**Significance**: The interactive parametric estimation problem considered by the paper is an interesting one and is directly relevant to the community.

**Weaknesses:**

Overall, the paper itself is quite nicely written and the topic is interesting. Some comments regarding weaknesses include:

1. The paper is somewhat structured reversely in many scenarios, where the paper references equations that appear later on in the paper. This may cause a little bit of confusion for readers.

2. It may be helpful to highlight the difference in technical contributions (for example, proof techniques) between this work, [4], and [16], considering the problem setup is quite similar and both this paper and [16] "build upon the framework presented for the discrete setting in [4]". What are some scenarios where the minimax results in this paper can significantly improve upon results in [16]?

**Questions:**

1. While the paper focuses on *interactive* protocols, it is curious whether similar techniques can be applied to the general parametric estimation model (i.e., in the setting where Y = X) and achieve tighter lower bounds than previous works, considering the techniques used in this paper do not depend on "bounded ratio" type assumptions as claimed in the paper. Would it be possible to use results in the paper to achieve significant improvement to classic minimax bounds for sparse/non-sparse Gaussians with non-identity covariance $\mathbb{I}$ such as in the cases where covariance matrix eigenvalues are largely varying (i.e., on a different order)?

2. How much are we sacrificing by using the inequality in (30) within the supplementary material? Would not dropping the second term in the denominator give improved results?

3. How limiting is the assumption of Assumption 4 and its corresponding Lemma 1? In other words, how easy is it to find $\theta$'s that satisfy this assumption?

4. It may be beneficial to add some references to the parts of the paper where it is claimed that previous works regarding bounded ratio type assumptions are "flawed" (for example, maybe previous papers that analyze this flaw).


Some additional suggestions/typos/confusions:

a. The term $S$ is not clearly defined in Equation (2). It may help to add some description of what this means.

b. Line 193 seems to be incorrectly formatted.

**Limitations:**

There does not appear to be any negative societal impact associated with this paper.

---

> ### Author Rebuttal · Authors · 2023-08-10
>
> We thank the reviewer for the detailed reading and acknowledging the novelty and significance of the paper. Below we address technical questions raised in the review.
>
> #### ***It may be helpful to highlight the difference in technical contributions (for example, proof techniques) between this work, [4], and [16], ...***
> For comparison of our result to [4], see response to Reviewer DETb. While [16] does build on [4] as well, our goal and results are significantly more general: our objective was to obtain results under minimal assumptions, and in particular without any “bounded ratio assumption”. This is in particular apparent in the results of [16]:
>
> - Their Assumption 2 (“Likelihood ratio condition”) is exactly such a bounded-ratio assumption.
> - As a result, their bound for the sparse Gaussian case (Theorem 3 in their arXiv version: https://arxiv.org/abs/1802.08417) cannot use their general theorem (as the sparse case does not satisfy that assumption), and so they must use a completely different argument (Appendix D of their paper) which is restricted to the *non-interactive* case (termed simultaneous message passing protocols in their paper).
>
> This is to be contrasted to our general approach, which, free of that assumption, directly applies to sparse Gaussians in the interactive case as well (see lines 353-354, and Theorem 5 (Section G.2) of the supplemental of our paper)
>
> #### ***Would it be possible to use results in the paper to achieve significant improvement to classic minimax bounds ...***
> Our result can be viewed as an extension of classic methods for proving lower bounds of general parametric estimation problems to the case under information constraints in the interactive setting. Without information constraints, our technique will reduce back to these classic methods, e.g. Assouad’s method. In particular, the “bounded ratio” assumption we get rid of in this work is not needed without these information constraints since the divergence measures will have explicit forms when Y = X and the distribution of X is known.
>
> #### ***How much are we sacrificing by using the inequality in (30) within the supplementary material?***
> We drop that term to relate Hellinger distance to chi-square distance. It is critical for us, since we exploit the bilinear form of chi-square distance to get a handle over contraction in distances due to information constraints. In all the examples we have seen, this weakening does not hurt us. However, it is of course conceivable that there could be examples where this weakening of Hellinger distance to chi-square distance becomes the limitation.
>
> #### ***How limiting is the assumption of Assumption 4 and its corresponding Lemma 1?***
> This is a good question, with two (complementary) answers.
>
> - The first is that it is indeed limiting, analogously to the widespread Assouad’s lemma in standard minimax lower bounds: i.e., Assumption 4 does not really restrict further our results than it would using Assouad’s lemma in the “non-constrained” estimation literature.
> - The second is that it is not *really* limiting, in that this assumption (and corresponding lemma) is completely independent from Assumptions 1-3 and our main theorems. Namely, Theorems 1 and 2 (the key contribution of our work) allow to upper bound a given quantity, the average discrepancy (AD), and do not rely on Assumption 4 in any way. Then, Lemma 1 provides a lower bound on this same quantity (AD) (and does not rely on assumptions 1-3): putting the two together gives the minimax lower bounds. But if one found another, different way to lower bound (AD) that doesn’t need Assumption 4 (e.g., via a Fano-type argument instead of Assouad-type), or anything else really, then the same template would go through and one would readily get a minimax lower bound by combining this with Theorem 1 or 2. While we do not provide an alternative to Assumption 4/Lemma 1 in our paper (as this Assouad-like bound suffices for our purposes), such an alternative is entirely conceivable.
>
> #### ***It may be beneficial to add some references to the parts of the paper where it is claimed that previous works regarding bounded ratio type assumptions are "flawed"***
> This is a good suggestion, and we will do our best to incorporate it in the final version. Yet, as several of these flaws were confirmed via personal communication with the authors, we felt it would be awkward (and potentially in violation of the double-blind policy) to include them in the submission.

---

> > ### Comment · Reviewer_Zd1o · 2023-08-18
> > **Reply to rebuttal**
> >
> > Thank you for addressing my questions. Please add the discussions here into further revisions as appropriate.

---

### Official Review · Reviewer_VD3C · 2023-07-03

**Soundness:** 3 good
**Presentation:** 3 good
**Contribution:** 3 good
**Rating:** 7
**Confidence:** 2

**Summary:**

This work investigates the distributed parameter estimation problem under local information constraints such as communication constraints, local privacy constraints, and restricted measurements. The authors focus on information-theoretic lower bounds for the minimax error rates of these problems and present "plug-and-play" lower bounds that can be applied to various estimation problems. In addition, for most cases, the lower bounds are complemented with matching upper bounds.

**Strengths:**

This work presents a unified framework for deriving a variety of minimax lower bounds for various families of distributions under local information constraints for the distributed parameter estimation problem. Additionally, matching upper bounds have been provided to demonstrate that the lower bounds are tight for most settings.

**Weaknesses:**

It would be beneficial to see a table displaying the lower/upper bounds for the various cases considered in this work.

**Questions:**

no

---

> ### Author Rebuttal · Authors · 2023-08-10
>
> We thank the reviewer for the positive comments. We will improve our paper based on the suggestions in future revisions.

---

### Official Review · Reviewer_7efN · 2023-07-05

**Soundness:** 3 good
**Presentation:** 3 good
**Contribution:** 3 good
**Rating:** 7
**Confidence:** 2

**Summary:**

This paper studies parameter estimation problem in a setting with the following two important assumptions:
1. Raw samples X_i's are generated from a certain parametric distribution, however, they are not directly observed. Instead, Y_i's which are generated from channels subject to certain constraints are observed. These constraints can encapsulate various problems of interest, notably communication constraints and local privacy.
2. Interaction is allowed. That is, Y_i's are generated with memory.

The main result is a general abstract upper bound on the sensitivity of the transcript (i.e., public view Y_i's and common randomness if any) of the interactive protocol.
This, upon nontrivial specification, yields a suite of tight or nearly tight impossibility results for problems mentioned above.

**Strengths:**

I'm not an expert in this field.
Though the main text is somewhat dense, I found it pretty accessible.
Most importantly, the results seem pretty satisfactory.

**Weaknesses:**

I don't see obvious weaknesses.
Please instead see my questions below (most of which are very minor).

**Questions:**

Major questions:
1. Is it possible to explain why in the main bounds (Theorem 1 & 2), TV distance was used? Is this the only reasonable choice? If so, why?
If not, is using other distance/info measure beneficial in any way, e.g., simplifying the proof, improving leading constant factors, etc.?

2. I saw that most examples in this paper are regarding product (or "product-like") distributions (e.g., product binary distribution, isotropic Gaussian). And my shallow understanding is that this seems to be a limitation of the technique.
This question may go beyond the scope of the paper, but I'm wondering if it is possible to obtain similar results when certain structured correlation is present.
E.g., Gaussian with general covariance, or simple temporal correlation such as Markovian structure.

Other comments:

1. Line 28, "local information constraint". I'm not familiar with this line of work and this may be a silly question. Could the authors offer some examples of non-local/global constraints?

2. I feel like already in line 44, the buzz word "Assouad" should have been mentioned. This is a rather standard tool and a general audience can smell that when reading upon line 44.

3. Mysteriously the line breaks at 193.

4. In Equation (2), the Gothic letter under the first sup was defined in footnote 2. It took me a while to find it. Perhaps upgrade this footnote to the main text?

5. To information theorists the notation y^t (perhaps more commonly: y_1^t) for (y_1, ..., y_t) is common but I'm not sure how standard it is to a general reader, especially given that this can be confused with the t-th power of a number.
I encourage the authors define this notation somewhere.

6. The notation p_Z^{Y^n} denotes the density of (Y^n, U), not just Y^n. Somehow U is dropped. Is there a reason for doing so?

7. In Equation (3), the integral is against y w.r.t. mu. Sometimes it might be slightly more clear to explicitly write $\mathrm{d}\\, \mu(y)$.
Also, this measure mu was defined in footnote 2, which is somewhat hidden.

8. A typo on line 359, double commas following "i.e.".

---

> ### Author Rebuttal · Authors · 2023-08-10
>
> We thank the reviewer for their detailed reading and suggestions on improving the presentation of the paper. Below we address technical concerns raised by the reviewer.
>
> #### ***Is it possible to explain why in the main bounds (Theorem 1 & 2), TV distance was used? Is this the only reasonable choice?***
> Thanks for the great question. Here we discuss three commonly used discrepancy measures, including TV distance, Hellinger squared distance ($d_H$), and mutual information (expected KL divergence). In general, it holds that $d_{TV}^2 \le d_H^2 \le d_{KL}$. Our main bounds also hold if we switch to $d_H^2$, which is also how Theorem 1 is approved (see Appendix E.1). Further relaxing discrepancy measures to mutual information (or KL) is trickier. In particular, [16] (as referred in our paper) uses mutual information as their discrepancy measure, and as a result, their bound would require an additional “bounded ratio” assumption in the distribution family, which fails for the sparse Gaussian family (see details in the response to reviewer Zd1o regarding the comparison to [16]). To summarize, both $d_{TV}^2$ and $d_H^2$ could be used, and would lead to similar results, comparable both in terms of proof length and resulting constants (which, for the sake of this paper, we didn’t focus on optimizing). TV distance was chosen here mainly because of its convenience and more widespread use.
>
> #### ***Examples of "non-local constraints."***
> The central version of differential privacy [Dwork and Roth 14] requires that a private algorithm should output similar outputs for neighboring datasets (those differ by at most one data point). The constraint cannot be expressed as a local constraint to the best of our knowledge.
>
> Another type of “non-local constraint” would be adversarial corruptions of data (see, e.g., https://sites.google.com/view/ars-book/): while one can model the Huber contamination model as a local constraint (as each data point is corrupted in an i.i.d. fashion), the common “strong contamination model” which allows an adversary to corrupt say 10% of the samples (of its choice) in an arbitrary fashion is not a local constraint, as it allows for arbitrary correlations among corrupted samples.
>
> #### ***... if it is possible to obtain similar results when certain structured correlation is present. E.g., Gaussian with general covariance ...***
> See answer in the global response.
>
> [Dwork and Roth 14] Dwork, Cynthia, and Aaron Roth. "The algorithmic foundations of differential privacy." Foundations and Trends in Theoretical Computer Science 9.3–4 (2014): 211-407.

---

### Official Review · Reviewer_DETb · 2023-07-06

**Soundness:** 3 good
**Presentation:** 3 good
**Contribution:** 3 good
**Rating:** 6
**Confidence:** 4

**Summary:**

This paper studies distributed parameter estimation under local information constraints. In this problem, independent samples $X_1,..., X_n$ are generated from an unknown distribution $p_\theta$ from a parametric family $P_\Theta$ of distributions. The samples are not accessible directly, rather available is only partial inofmration $Y_1, ..., Y_n$  generated by passing the original samples through classical channels $W_i$. The goal is to estimate the underlying parameter $\theta$ by accessing $Y_1, ..., Y_n$. The channels can be interactive, dependig on the past observations. They also local information constraints,  including communication constraints, and privacy constraints.

This paper derives lower bounds on the sample compelxity of this problem in a general framework. Moreover, this work explores applications of the general bound to problems of high-dimensional mean estimation and distribution estimation, under privacy and communication constraints, for the entire family of $l_p$ loss functions for $p \geq 1$.

**Strengths:**

The problem formulation and the related results are substantially general and can be used in various problems involving different constraints.
The paper is rigorous and well written.

**Weaknesses:**

It is not clear how significant the generalization is compared to previous works (i.e., reference [4]). Is the paper generalizing [4] to $l_p$ loss functions? If that is the case, I am not sure how important that is beyond a mathematical curiosity.

Minor comment: it looks like that the symbol $\wedge$ is not defined.

**Questions:**

As another point, the current approach focuses on a finite perturbation subset of $\Theta$. How does this approach work for continuous parameter space $\Theta$ ? Do we need to make sure that the Z space is a good representation of $\Theta$?

**Limitations:**

See weaknesses.

---

> ### Author Rebuttal · Authors · 2023-08-09
>
> We thank the reviewer for the detailed reading and acknowledging the generality of our technique. Below we address the questions raised by the reviewer.
>
> #### ***The significance of the generalization compared to [4]***
> We stress that the results and techniques of [4] are specific to discrete distributions: extending these techniques to handle general distribution classes (e.g., high dimensional distributions) is a non-trivial task. For example, for high-dimensional distributions, to get the correct dependence of $d/\ell$ under communication constraints, we use a measure change bound (Lemma 3 in the appendix) to prove an upper bound of $\log|\mathcal{Y}|$ instead of $|\mathcal{Y}|$ in the information contraction bound (see Corollary 2). For discrete distributions, the latter will be sufficient. Moreover, the proof of the result in [4] relies on the sum of mutual information as a key step, which will inherently cause an “bounded ratio” assumption in the distribution family similar to [16]  (see details in the response to reviewer Zd1o regarding comparison to [16]). We resort to average TV distance as the discrepancy measure to remove this assumption. This being said, [4] also provides bounds for *testing* (and not just estimation), which our paper doesn’t include.
>
> As such, our paper is a considerable generalization of [4], going way beyond the simple extension to other $L_p$ norms.
>
> #### ***How does this approach work for continuous parameter space? Do we need to make sure that the $Z$ space is a good representation of $\Theta$?***
>
> Yes, indeed. Making sure that the $Z$ space captures the hardness of $\Theta$ is an important step for using our framework to prove information-theoretic lower bounds. This typically requires an understanding of the structure of the parametric estimation problem, which is also necessary for proving lower bounds for parameter estimation in classic settings without local information constraints. Our framework captures the additional cost imposed by local information constraints on top of this.

---

> > ### Comment · Reviewer_DETb · 2023-08-18
> >
> > Thank you for the rebuttal. Some of my concerns are addressed, and I increase my score.

---

### Official Review · Reviewer_D6mk · 2023-07-26

**Soundness:** 4 excellent
**Presentation:** 4 excellent
**Contribution:** 3 good
**Rating:** 8
**Confidence:** 4

**Summary:**

The work considers the problem of proving sample complexity lower bounds for parameter/density estimation in a distributed setting with information constraints (local privacy, communication constraints etc). What distinguishes this paper from prior work is that it focuses on the scenario where there is interaction between the users who share their data. Thus, while the data-points are initially generated in an i.i.d. fashion, when the time comes for the data-points to be subjected to the information constraints, the results depend on previous users' actions.

The authors develop general tools for obtaining lower bounds in this setting, where the main challenge involves dealing with the dependencies implied by the interactions between users. The main tool developed is a lemma that is essentially reminiscent of Assouad's method that is adapted to the setting of sequentially interactive distributed protocols with information constraints. The resulting framework manages to capture various existing techniques such as strong data-processing inequalities and bounds derived using the distributed version of the Bayesian Cramer-Rao bound know as the Van Trees inequality. The paper also includes applications of these techniques to a number of estimation problems, while the derived lower bounds are in most cases complemented by nearly-matching upper bounds. Finally, there's a description of how the techniques could be adapted to work in the fully interactive setting as well.

**Strengths:**

This is a quite strong paper with technical contributions that improve our understanding of statistical estimation tasks in the distributed setting. The work provides both techniques (which are presented in a quite general way) and applications (which confirm the effectiveness of said techniques). The applications include the fundamental tasks of mean estimation for binary product distributions and Gaussian distributions under the $\ell_p$-norm, as well as density estimation for discrete distributions. The work is also notable for identifying mistakes in previous papers where some of these results were claimed and using the developed framework to fix the proofs. Finally, the writing is very clear and the work is clearly positioned within the overall literature.

**Weaknesses:**

No obvious weaknesses.

**Questions:**

I have one question. The applications given in this paper within the context of parameter estimation for high-dimensional distributions all involve independent marginals (binary product distributions, spherical Gaussians). Additionally, the lemma the proofs of these statements rely on (lemma 3) is an analogue of Assouad's method. Assouad's method assumes that the family of distributions considered as part of proving the lower bound are parameterized by the vertices of the boolean hypercube, with each coordinate being drawn independently. This frequently ends up limiting the applicability of the method to problems involving distributions with independent marginals. Is this a limitation of the work, or is there evidence the method can lead to tight lower bounds even with non-product distributions (e.g., for Gaussian covariance estimation)?

**Limitations:**

This is a purely theoretical work with a focus on lower bounds. Thus, there is no apparent societal impact (positive or negative).

---

> ### Author Rebuttal · Authors · 2023-08-10
>
> We thank the reviewer for the very positive assessment of our work. We refer the reviewer to the global response for our discussion on generalizing the technique beyond distributions with independent marginals.

---

> > ### Comment · Reviewer_D6mk · 2023-08-18
> >
> > Thank you very much! My score remains unchanged.

---

### Author Rebuttal · Authors · 2023-08-09

We thank the reviewers for their careful reading of our submission, and are delighted to see their very positive assessments of our work. We will take into account their comments and suggestions in the final version of our paper, and respond individually to their questions below. We first address a question mentioned by several reviewers.

#### ***Whether our techniques could apply beyond identity-covariance matrix or independent coordinates.***

This is, indeed, a very natural question: while we believe our techniques can provide lower bounds for this case (for a suitable family of hard instances, such as the one described in Lemma 6.11 of [https://arxiv.org/abs/1805.00216 ], which is indeed parameterized by a vector of $\binom{d}{2}$ independent $\pm 1$ parameters), such a lower bound would most likely be quite technical, and lengthen the paper significantly. As our main focus here was to establish the general lower bound framework and demonstrate its applicability via a few notable examples, we left these additional applications for future work.

---

### Decision · Program_Chairs · 2023-09-21

**Decision:**

Accept (poster)

**Comment:**

The authors propose a framework for proving lower bounds for high dimensional estimation in the interactive setting with communication constraints.

 The reviewers found the authors contributions in finding a tight and unifying approach (which recovers some well-known lower bounds) significant and well presented. I therefore recommend acceptance.